# E(3)-EQUIVARIANT MODELS CANNOT LEARN CHIRALITY: FIELD-BASED MOLECULAR GENERATION

**Alexandru Dumitrescu**[†]**, Dani Korpela**[†]**, Markus Heinonen**[†]**, Yogesh Verma**[†]**,**
**Valerii Iakovlev**[†]**, Vikas Garg**[†,‡] **& Harri Lähdesmäki**[†]
[†]Department of Computer Science, Aalto University
[‡]YaiYai Ltd
Correspondence: {alexandru.dumitrescu}@aalto.fi

## ABSTRACT

Obtaining the desired effect of drugs is highly dependent on their molecular geometries. Thus, the current prevailing paradigm focuses on 3D point-cloud atom representations, utilizing graph neural network (GNN) parametrizations, with rotational symmetries baked in via E(3) invariant layers. We prove that such models must necessarily disregard chirality, a geometric property of the molecules that cannot be superimposed on their mirror image by rotation and translation. Chirality plays a key role in determining drug safety and potency. To address this glaring issue, we introduce a novel field-based representation, proposing reference rotations that replace rotational symmetry constraints. The proposed model captures all molecular geometries including chirality, while still achieving highly competitive performance with E(3)-based methods across standard benchmarking metrics. Code is available at https://dumitrescu-alexandru.github.io/FMG-web/.

## 1 INTRODUCTION

Developing generative models for specific data modalities requires carefully choosing the generative method, feature representation, and NN architecture. Field-based representations may offer a unifying perspective, extending the usage of highly successful architectures from natural language or images to other input modalities Zhuang et al. (2023). We develop Field-based Molecule Generation (FMG), a generative model of three-dimensional field representations of molecules, which encode the joint distribution of molecular graphs and 3D conformations.

Organic molecules generally contain a restricted number of atom types, bound mostly through covalent bonds, and they can be used as drugs, inhibiting or promoting biological processes. Their effectiveness crucially depends on successfully binding and interacting with proteins or other small molecules in the organism. In turn, the interaction between two molecules can only be determined once their 3D conformations are known. This motivated the development of many generative models for molecular conformations. Recently, the vast majority of work has been done on generative model parameterizations that are invariant to the Euclidean group of isometries in 3D, E(3) (Hoogeboom et al., 2022; Peng et al., 2023; Vignac et al., 2023b; Xu et al., 2023; Wu et al., 2022).

The E(3) group contains any reflections, rotations, and translations, whereas SE(3) ⊂ E(3) (special E(3)) contains only proper rotations and translations. In general, molecules are not E(3) invariant. Certain molecules are sensitive to reflections meaning that one cannot superimpose the molecule to its mirror-image by (proper) rotations and translations. This property is called chirality or handedness, and chiral molecules form enantiomer pairs (Figure 1).

Chiral centers are often tetrahedral carbon configurations containing four unique groups. In molecular biology, chirality is crucial to the potency and safety of drugs. In ibuprofen, albuterol, and naproxen, one of the two enantiomers was shown to be more effective, while in thalidomide and propoxyphene, the incorrect enantiomer is toxic and can lead to severe side effects (Smith, 2009). In proteins, out of 21 standard amino acids, only Glycine is not chiral. Although classically it was believed that only one chiral configuration is present in nature (Devínsky, 2021), patients with disorders such as cataracts, macular degeneration, arteriosclerosis, and Alzheimer's disease present both

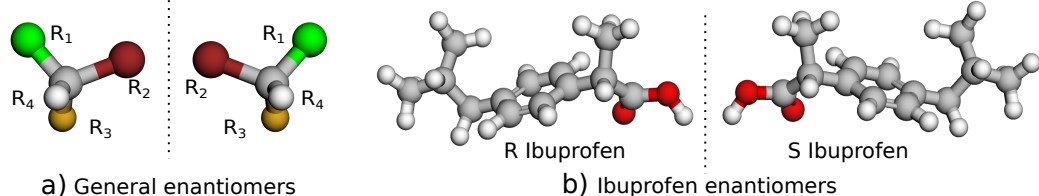

a) General enantiomers  b) Ibuprofen enantiomers

Figure 1: **E(3) invariant spatial features (e.g., relative bond angles and distances) are not sufficient to represent chirality.** Subfigure a) depicts a general chiral molecule pair, which cannot be superimposed by rotations. b) Chiral S-ibuprofen is an efficient COX-inhibitor resulting in pain and inflammation relief, while the mirror R-ibuprofen is not (Evans, 2001).

(Masters et al., 1977; Fujii et al., 2018). In the drug-like molecule dataset, GEOM, 27% of the molecules contain chiral centers.

In essence, reflections are distance preserving, and therefore parameterizations that utilize Euclidean distances will be unable to distinguish enantiomers. Later on, we prove theoretical results that show how designing enantiomer (reflection) variant features (SE(3) invariant) grows the complexity from $\mathcal{O}(n^2)$ (for pairwise Euclidean distances between $N$ atoms) to $\mathcal{O}(n^4)$. To account for chiral centers, we instead chose to replace rotation symmetry constraints, and propose reference rotations, which bring our performance to highly competitive results with E(3) invariant methods.

We organize our contributions as follows:

**Theoretical:** We prove that functions of E(3) invariant features fail to properly model chiral molecules. We then link the parametrizing function's E(3) invariance to the resulting generative E(3) equivariance. Finally, we prove that SE(3)-invariant, chiral aware features can only be functions of a least four $\mathbb{R}^3$ vectors, and the number of such features scales in $\mathcal{O}(n^4)$.

**Practical:** We provide solutions to the challenges of handling 3D fields. We present a method of reorienting molecules to deterministic references, introduce bond fields, describe our diffusion process design, and complete our pipeline with molecular graph optimization based on fields.

**Empirical:** Put together, our technical contributions result in SOTA or highly competitive performance compared to point-cloud approaches, while keeping chiral-awareness.

## 2 RELATED WORKS

A large number of generative models for molecules have been proposed. We distinguish them primarily by their molecular representation instead of generative method families.

**SMILES** SMILES is a string-based representation in which molecular graphs can be sufficiently encoded (Weininger, 1988), and has been employed in autoregressive (Segler et al., 2018) and variational autoencoder (VAE) models (Gómez-Bombarelli et al., 2018; Kusner et al., 2017). An important limitation of SMILES used as input representations for machine learning is that similar molecular graphs may have very different SMILES encodings (Jin et al., 2018).

**Graphs** Alternative molecule representations consist of molecular graphs, where vertices encode atom types and their covalent bonds are represented through adjacency matrices. VAEs (Liu et al., 2018; Jin et al., 2018), flow-based models (Shi et al., 2020; Verma et al., 2022), reinforcement learning (Shi et al., 2020; You et al., 2018; Zhou et al., 2019) and diffusion-based methods (Vignac et al., 2023a; Mercatali et al., 2024) have been developed. The main drawback of only generating molecular graphs is that full molecular geometry specification is missing, and the functionality of various drug-like molecules depends on it.

**Point clouds** The three-dimensional configuration of a molecule can be encoded as $\mathbb{R}^3$ point clouds of atoms. Molecules reside in the $\mathbb{R}^3$ space, and their conformations are symmetric w.r.t. global translations and rotations. Inspired by this, E(3) equivariant generative methods have been developed. Autoregressive models (Gebauer et al., 2018; 2019), GNN parametrized diffusion (Hoogeboom et al., 2022; Peng et al., 2023; Vignac et al., 2023b; Xu et al., 2023; Wu et al., 2022), and transformer parametrized diffusion (Hua et al., 2023; Huang et al., 2024) methods have recently gained a lot of interest. Although rotational invariance is well motivated, these methods also encode the undesired invariance to reflections, which are part of the E(3) group.

**Fields** The $\mathbb{R}^3$ coordinates of atoms in molecules can be used as the mean positions of e.g. Gaussians, which fill the three-dimensional space. These densities can be interpreted as fields, where each molecule is represented by a function on $\mathbb{R}^3$, with values given by the densities of the atoms. We crucially distinguish this approach from the previously developed methods that use point cloud formulations by the E(3) symmetries they adhere to. Namely, our approach only respects translation invariance through the CNN inductive biases, while point cloud approaches respect all E(3) symmetries. At the cost of losing rotational invariance, our method can generate chiral conformations according to data statistics, while E(3) invariant formulations cannot. Similar input representations have been previously modeled using VAE (Ragoza et al., 2020) and score-based models (Pinheiro et al., 2023). The latter is the closest to our work, which bases its method on the neural empirical Bayes framework and only generates atom voxels, based on which they use post-generation bond extraction. Using diffusion models, we show that field representations can reach SOTA performance while generating the full molecule specification (including bond fields).

**Chiral-aware methods** Prior, chiral-aware discriminative methods for molecular analysis have been developed in (Adams et al., 2022; Liu et al., 2022). Jing et al. (2022) also accounts for chirality in a generative setting using a diffusion model, but considers the conditional generation of 3D molecular conformations given a molecular graph $p(C|G)$, while the scope of this work is on the joint generation $p(C, G)$. Finally, Luo & Ji (2022) uses (Liu et al., 2022) NN parametrization of a joint $p(C, G)$ AR generative model that, in principle, is chiral-aware, but does not discuss chirality. AR methods need to define arbitrary node order generation, and whether they can reach diffusion model performance despite this fact remains an open question. In general, chiral-aware SE(3) diffusion models on point clouds are surprisingly difficult to develop.

## 3 CHIRALITY AND ISOMETRY GROUPS

Most recent work on molecule generation model E(3) equivariant distributions. Although molecules respect rotation and translation symmetries, chiral molecules are not symmetric to reflections. In such cases, the reflected version of a therapeutic drug can become toxic (Smith, 2009).

**Proposition 1.** *If $p_\phi$ is an E(3) invariant probability distribution, then $p_\phi(\mathbf{m}) = p_\phi(\mathbf{m}')$, where $\mathbf{m}$ and $\mathbf{m}'$ are the $\mathbb{R}^{3 \times N}$ positions of enantiomer pairs of molecules with $N$ atoms.*

Proposition 1 is a consequence of the fact that $p_\phi(\mathbf{m}) = p_\phi(\mathbf{Rm}), \forall\, \mathbf{R}$ orthogonal, $\det \mathbf{R} = \pm 1$. This implies that NN parameterizations of point-cloud diffusion models should be SE(3) invariant, but we argue that this becomes intractable.

**Lemma 2.** *Let $f : \mathbb{R}^{m \times n} \to \mathcal{S}$, for some set $\mathcal{S}$, be a function of $m$ $n$-dimensional vectors. If $m \le n$ and $f$ is SE(n) invariant, then $f$ is E(n) invariant.*

For example, in $\mathbb{R}^3$, the reflection of a planar object ($m = n = 3$) can be recovered by rotations and translations. This implies that there is no SE(3) invariant function that, applied on triplets of atom positions, would distinguish enantiomer pairs.

**Proposition 3.** *Let $n$ be the number of atoms of a molecule. The number of features that encode all chiral configurations of random point clouds scales in $O(n^4)$.*

The result implies that SE(3) (and not E(3)) invariant features scale prohibitively with the number of atoms.

Proofs can be found in Appendix B. There, we also concretely describe the model types we refer to in Proposition 3, which include all SOTA diffusion models (Hoogeboom et al., 2022; Peng et al., 2023; Vignac et al., 2023b; Xu et al., 2023; Wu et al., 2022).

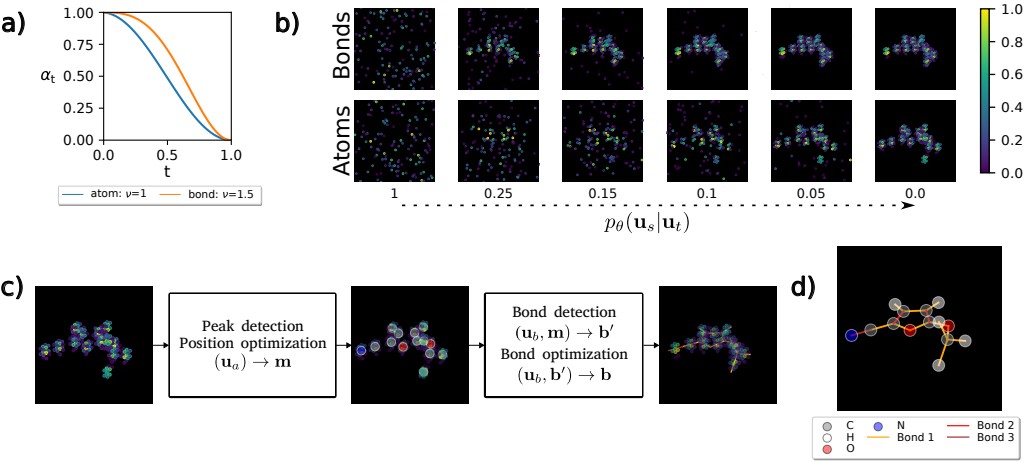

Figure 2: a) Atom $\mathbf{u}_a$ and bond $\mathbf{u}_b$ fields noise schedules. b) Sampling illustration. A subset of 200 3D locations with the highest values are shown for all $\mathbf{u}_a$ and $\mathbf{u}_b$ channels. c) Extracting atoms and bonds from the resulting $\mathbf{u}_a$ and $\mathbf{u}_b$ fields at $t = 0$. d) Visualizing the optimized atoms and bonds.

## 4 MOLECULAR FIELD DIFFUSION

In this Section, we propose a diffusion model to generate molecular fields.

### 4.1 FIELD REPRESENTATION

We consider a molecule as a set of atoms $\{a_i, \mathbf{m}_i\}$ and a set of bonds $\{b_{ij}, \mathbf{m}_{ij}\}$, with atom types $a_i \in \mathcal{A} = \{\mathrm{H}, \mathrm{C}, \mathrm{N}, \mathrm{O}, \mathrm{P}, \mathrm{S}, \ldots\}$, bond types $b_{ij} \in \mathcal{B} = \{b_1, b_2, b_3\}$, and we assume atom locations $\mathbf{m}_i \in \mathbb{R}^3$ and mid-point bond locations $\mathbf{m}_{ij} = \frac{\mathbf{m}_i + \mathbf{m}_j}{2}$ in Ångströms.

We start by representing a molecule using a collection of continuous 3D fields, one for each atom and bond type. We define the field for atom type $a$, $u_a$, as a mixture of radial basis functions (RBFs) positioned on the atoms' centers

$$u_a(\mathbf{x}) = \gamma_a \sum_{i=1}^{N} \mathbb{1}[a_i = a] \exp\left(-\frac{1}{2\sigma_a^2} ||\mathbf{x} - \mathbf{m}_i||^2\right), \tag{1}$$

where $N$ is the number of atoms, $\mathbb{1}[\cdot]$ is the indicator function, the variance $\sigma_a^2$ represents the size of atom type $a$, and $\gamma_a$ normalises the channels to unit interval $u_a(\mathbf{x}) \in [0, 1]$ (see Appendix C for field parameter values). We create bond fields similarly

$$u_b(\mathbf{x}) = \gamma_b \sum_{(i,j)} \mathbb{1}[b_{ij} = b] \exp\left(-\frac{1}{2\sigma_b^2} ||\mathbf{x} - \mathbf{m}_{ij}||^2\right). \tag{2}$$

We evaluate a molecule's atom and bond fields $u_a, u_b$ at a fixed set $\mathcal{G}$ of $H \times W \times D$ (height, width, and depth) locations, and create a vector-valued field based on which multi-channel 4D tensors are derived

$$\mathbf{u} = (\mathbf{u}_{\mathrm{H}}, \mathbf{u}_{\mathrm{C}}, \ldots; \mathbf{u}_{b_1}, \mathbf{u}_{b_2}, \mathbf{u}_{b_3}) \in \mathbb{R}^{K \times H \times W \times D}, \tag{3}$$

where $K = |\mathcal{A}| + |\mathcal{B}|$ is the number of channels. We use tensor $\mathbf{u}$ as the input molecule representation for the diffusion model.

### 4.2 DIFFUSION MODEL

We consider a diffusion generative model to generate molecular fields $\mathbf{u}$. In diffusion models, the data is incrementally noised by a forward process, and our goal is to learn a denoising reverse process

parameterised by $\theta$. We consider the field tensors $\mathbf{u}_t$ over time $t \in [0, 1]$, where $\mathbf{u}_0$ at time $t = 0$ are the observed or generated tensors. Figure 2b depicts an example of our denoising reverse process.

We follow the denoising diffusion probabilistic model (DDPM) proposed in (Sohl-Dickstein et al., 2015; Ho et al., 2020; Kingma et al., 2021), and refer the reader to the original publications for full derivations. We note that our tensor fields are effectively multi-channel 3D images, and thus image-based diffusion models translate to molecular voxel grid fields in a straightforward manner.

**Forward process** We begin by assuming a Gaussian forward process

$$q(\mathbf{u}_t|\mathbf{u}_0) = \mathcal{N}(\mathbf{u}_t|\alpha_t\mathbf{u}_0, \sigma_t^2\mathbf{I}), \tag{4}$$

where $\alpha_t \in [0, 1]$ monotonically reduces the original signal $\mathbf{u}_0$, while $\sigma_t \in [0, 1]$ monotonically increases noise. We choose a variance-preserving noise specification, where

$$\alpha_t = \sqrt{1 - \sigma_t^2} \tag{5}$$

At $t = 1$ we obtain pure noise $q(\mathbf{u}_1|\mathbf{u}_0) = \mathcal{N}(\mathbf{0}, \mathbf{I})$.

**Forward posterior** The forward process admits a tractable conditional posterior distribution for an intermediate state $\mathbf{u}_s$ if we know both the original state $\mathbf{u}_0$ and further noised state $\mathbf{u}_t$ for $s < t$,

$$q(\mathbf{u}_s|\mathbf{u}_t, \mathbf{u}_0) = \mathcal{N}\left(\frac{\alpha_{ts}\sigma_s^2}{\sigma_t^2}\mathbf{u}_t + \frac{\alpha_s\sigma_{ts}^2}{\sigma_t^2}\mathbf{u}_0, \frac{\sigma_{ts}^2\sigma_s^2}{\sigma_t^2}\mathbf{I}\right), \tag{6}$$

where $\alpha_{ts} = \alpha_t/\alpha_s$ and $\sigma_{ts}^2 = \sigma_t^2 - \alpha_{ts}^2\sigma_s^2$ (See cf. Kingma et al. (2021)). Due to Gaussianity of the forward, the posterior interpolates linearly between the endpoints while adding variance.

**Reverse process** Our goal is to learn a generative process that reverses the forward by matching their marginals. The reverse process can be framed as the posterior of the forward

$$p_\theta(\mathbf{u}_s|\mathbf{u}_t) = q\left(\mathbf{u}_s\big|\mathbf{u}_t, \mathbf{u}_0 := \hat{\mathbf{u}}_\theta(\mathbf{u}_t, t)\right), \tag{7}$$

where we replace the observation conditioning $\mathbf{u}_0$ with a fully denoised prediction $\hat{\mathbf{u}}_\theta(\mathbf{u}_t, t)$ from a neural network parameterised by $\theta$.

**Parameterisation** In practice, it is often more efficient to train a neural network to predict the added noise $\boldsymbol{\epsilon} \sim \mathcal{N}(\mathbf{0}, \mathbf{I})$ of

$$\mathbf{u}_t = \alpha_t\mathbf{u}_0 + \sigma_t\boldsymbol{\epsilon}, \tag{8}$$

than to predict the signal $\mathbf{u}_0$. We then train a neural network $\boldsymbol{\epsilon}_\theta(\mathbf{u}_t)$ to predict the noise $\boldsymbol{\epsilon}$, and follow the above equations with a substitution

$$\hat{\mathbf{u}}_\theta = \mathbf{u}_t/\alpha_t - \sigma_t\hat{\boldsymbol{\epsilon}}_\theta(\mathbf{u}_t, t)/\alpha_t. \tag{9}$$

**Loss** The diffusion model admits a variational lower bound (Kingma et al., 2021), but earlier works have argued for a simplified loss (Ho et al., 2020)

$$\mathcal{L}_{\text{simple}} = \mathbb{E}_{\mathbf{u}_0, t, \boldsymbol{\epsilon}}\left[||\boldsymbol{\epsilon} - \hat{\boldsymbol{\epsilon}}_\theta(\mathbf{u}_t, t)||^2\right], \tag{10}$$

where we learn to denoise arbitrary combinations of observations $\mathbf{u}_0$, noises $\boldsymbol{\epsilon}$ and noise scales $t \sim U(0, 1)$. We also opt for the simple loss $\mathcal{L}_{\text{simple}}$.

## 4.3 NOISE SCHEDULES

To further emphasize the stability of generated molecules, Vignac et al. (2023b) and Peng et al. (2023) employed separate noise schedulers for the bonds, atom types, and their positions using a modified cosine and a sigmoid scheduler, respectively. We also observed increased performance using a similar approach to Vignac et al. (2023b), which adds a parameter $\nu$ to the original cosine schedule from (Nichol & Dhariwal, 2021):

$$\alpha_t = \cos\left(\frac{\pi}{2}\frac{(t + s)^\nu}{1 + s}\right)^2, \tag{11}$$

where $\alpha_t$ determines the signal amplitude for each time $t$ in Equation 4, and we set $\nu$ to 1 and 1.5 for atom and bond channels, respectively. The schedule $\alpha_t$ and its effect on generation can be seen in Figure 2, where bonds are denoised faster than atoms, increasing the generated molecules' stability.

## 4.4 CLASSIFIER FREE GUIDANCE

Ho & Salimans (2022) empirically showed that using an approximation of the implicit classifier $p(\mathbf{y}|\mathbf{u}_t) \propto p(\mathbf{u}_t|\mathbf{y})/p(\mathbf{u}_t)$ achieves similar conditional generation performance to (Dhariwal & Nichol, 2021), without needing a separately trained classifier $p_\phi(\mathbf{y}|\mathbf{u}_t)$ for multiple noise levels $t$. During sampling, the conditional and unconditional noise estimators $\boldsymbol{\epsilon}_\theta(\mathbf{u}_t; \mathbf{y})$ and $\boldsymbol{\epsilon}_\theta(\mathbf{u}_t)$ are linearly interpolated:

$$\tilde{\boldsymbol{\epsilon}}_\theta(\mathbf{u}_t; \mathbf{y}) = (1 + \beta)\boldsymbol{\epsilon}_\theta(\mathbf{u}_t; \mathbf{y}) - \beta\boldsymbol{\epsilon}_\theta(\mathbf{u}_t), \tag{12}$$

where we note that $\boldsymbol{\epsilon}_\theta(\mathbf{u}_t; \mathbf{y})$ and $\boldsymbol{\epsilon}_\theta(\mathbf{u}_t)$ are up to a constant equal to the conditional and marginal score estimates $\nabla_\mathbf{u} \log p_\theta(\mathbf{u}_t|\mathbf{y})$ and $\nabla_\mathbf{u} \log p_\theta(\mathbf{u}_t)$.

Point-cloud approaches always strictly enforce the generated atom counts. We determine the effect of atom count conditioning using classifier-free guidance on generated molecule statistics. While not strictly necessary for our field-based formulation, we show that atom count conditioning yields molecule graph quality improvements. Further details can be found in Appendix G.

## 4.5 ARCHITECTURE CHOICE, ROTATION INVARIANCE AND SCALING

We parametrize the denoiser $\boldsymbol{\epsilon}_\theta$ with a three-dimensional U-Net architecture, which consists of ResNet blocks, scaling layers, and attention layers. CNN layers provide useful translational invariance inductive biases but lack rotation invariance.

We propose a simple technique that side-steps the issue by orienting each molecule wrt its principal components $\mathbf{V} = (\mathbf{v}_1, \mathbf{v}_2, \mathbf{v}_3)$ of their atom positions $\mathbf{M} = (\mathbf{m}_1, \ldots, \mathbf{m}_N)$. Oriented molecules $\tilde{\mathbf{M}} = \mathbf{V}^T\mathbf{M}$ have a canonical frame of reference $\mathbf{V}$, where the axis of the largest variation is placed on the horizontal axis, the second-largest on the vertical axis, and the smallest on the depth axis. This ensures a high overlap between the large values of fields from different molecules.

Compared to previous methods that use E(3) invariant NN and can generate molecules with any rotation, our method (on optimality) will only generate molecules according to this frame of reference. NN hyperparameters can be found in Appendix D.

Our method's scaling is dependent on the evaluated field locations $\mathcal{G}$. In Appendix J, we derive the upper limit of our method's scaling and show that it differs substantially from empirical values, even surpassing the generation speed of (Hoogeboom et al., 2022) for the largest molecules.

## 4.6 EXTRACTING MOLECULAR GRAPHS

We briefly illustrate the molecular graph extraction method and give additional details in Appendix E. To extract a molecular graph from the generated fields $\mathbf{u} = (\mathbf{u}_H, \mathbf{u}_C, \ldots, \mathbf{u}_{b_1}, \mathbf{u}_{b_2}, \mathbf{u}_{b_3})$, we first initialize the number of atoms $N$, their positions $\{\mathbf{m}_i\}$, and their types $\{\mathbf{a}_i\}$, using a computationally efficient peak picking algorithm which retrieves sets $M_a$ of atom positions $\mathbf{m}_i$, for each atom type. We then optimize the mean-squared error loss between the values of the generated fields $\mathbf{u}_a$, and the corresponding atom RBF function values determined by $\mathbf{m}_i \in M_a$ by matching the generated atom fields $\mathbf{u}_a$ against the RBF mixtures induced by $\{\mathbf{m}_i\}$ across the grid points $\mathcal{G}$

$$\underset{\{\mathbf{m}_i\}}{\arg\min} \sum_{a \in \mathcal{A}} \sum_{\mathbf{x} \in \mathcal{G}} \left[ \mathbf{u}_a(\mathbf{x}) - \gamma_a \sum_{i=1}^{N} \mathbb{1}[a_i = a] \exp\left(-\frac{1}{2\sigma_a^2}||\mathbf{x} - \mathbf{m}_i||^2\right) \right]^2. \tag{13}$$

Similarly, given the fixed, optimized atom positions $\{\mathbf{m}_i\}$ we initialize bonds. All atom position pairs $(\mathbf{m}_i, \mathbf{m}_j)$ within certain distances define our initial bond set $B$. We then consider sets $B_{ij,b}$ of values $\mathbf{u}_b$ around bond positions $\mathbf{m}_{ij} = (\mathbf{m}_i + \mathbf{m}_j)/2$, that are above predefined thresholds, for every $(\mathbf{m}_i, \mathbf{m}_j) \in B$. For all $B_{ij,b} \neq \varnothing$ we assign $\gamma_{ij}$ and optimize:

$$\underset{\gamma_{ij}}{\arg\min} \sum_{b \in \mathcal{B}} \sum_{\mathbf{x} \in \mathcal{G}} \left( \mathbf{u}_b(\mathbf{x}) - \sum_{(i,j)} \gamma_{ij} \mathbb{1}[b_{ij} = b] \exp\left(-\frac{1}{2\sigma_b^2}||\mathbf{x} - \mathbf{m}_{ij}||^2\right) \right)^2, \tag{14}$$

where the small RBF component values $\gamma_{ij}$ corresponding to erroneously detected bonds are removed.

## 5 Evaluation

We briefly present the benchmarking metrics and refer to Appendix H for details and definitions.

**Basic molecule properties**   We use atom and molecule neutrality and validity. Neutrality is the same metric defined in (Hoogeboom et al., 2022) as "stability", but, since other work considered stable atoms even those without a formal charge of 0 (Vignac et al., 2023b), we denote it neutrality, to clarify and distinguish from their definitions Atom neutrality is the percentage of atoms with a formal charge of 0, while we consider molecules as neutral if all their atoms are neutral (note how the definition slightly differs from the chemical one). Unlike validity, neutrality cannot be trivially increased by e.g., generating only covalent type 1 bonds, or other trivial solutions that may increase this metric at the expense of other distribution properties (Hoogeboom et al., 2022). We treat neutrality like any other distribution property, and we aim to match the generated neutral molecule fraction of the data.

**Molecule graph quality**   We compute atom total variation ($TV_a$) and bond total variation ($TV_b$) between the datasets and generated distributions, and report the sum of the distances over all atom and bond types, respectively.

Next, we report the average maximum similarity between the test dataset and generated molecules (MST). We compute the Morgan fingerprints (Zhong & Guan, 2023) of all the generated and test dataset molecular graphs and determine the maximum Tanimoto similarity between each generated molecule and the test set.

**Molecular conformation metrics**   To assess the quality of the generated 3D molecular conformations, we employ the Wasserstein metrics introduced in (Vignac et al., 2023b), and compute bond angle (BA) and bond length (BL) $W_1$ distance between generated and data distributions. We additionally compute conformational energies ($CE_{MMFF}$ and $CE_{xTB}$) $W_1$ distances, based on the MMFF and GFN2-xTB (Bannwarth et al., 2019) energies, respectively, of generated and data conformations. Since point-cloud methods may create large outliers and $W_1$ distances are outlier sensitive, we restrict generated angles and lengths to those generated within the data limits (see Appendix H).

## 6 Experiments and results

We report the basic molecule properties and molecule graph quality metrics in Tables 1 and 2, while molecule conformation and conditional generation results can be found in Appendix A.1 and A.2.

Previous methods have employed many different settings for molecule generation. While all of them may be valid approaches, it makes benchmarking challenging. We note in Appendix I how explicit aromatic generation makes it difficult to compute and compare certain metric values against Kekule representations (used in this work). In Tables 1 and 2, we annotate explicit aromatic formulations with (‡). Methods that employ post-generation corrections or algorithmic bond extractions are annotated with (†). The methods we tested ourselves are marked with (*).

### 6.1 QM9

QM9 (Ramakrishnan et al., 2014) is a small molecule dataset, containing 134k molecules of at most 29 atoms (including hydrogen). We use the splits from (Hoogeboom et al., 2022), with 100k, 18k, and 10k molecules for training, validation, and testing respectively. We use 98522 molecules out of 100k, selected s.t. $\mathbf{m}_i \pm 2\sigma$ of all atom RBF components fit within the evaluated field positions $\mathcal{G}$.

Basic molecular properties are reported in Table 1. We achieve state-of-the-art performance in terms of molecule and atom neutrality. In Table 2, the maximum test similarity and the bond distribution are competitive to the point-cloud approaches, while MiDi performs very well on $TV_a$.

This is likely an effect of our model trading the correct atom count generation for improved neutrality. Point cloud methods are hard-constrained to generate specific numbers of atoms, while our method may remove or add atoms in favor of other properties. In Appendix G, we report a small shift in the generated atom count distribution compared to the desired conditioning, while increasing the strength of classification guidance $\beta$ in Table 3 significantly decreases $TV_a$.

Table 1: Basic molecule properties for molecules generated by models trained on the QM9 and GEOM-Drugs datasets, based on 10k samples. We report the mean and standard deviation of the metrics for three different runs (each generating 10k samples) for QM9. The remark column (R) symbols represent: (*) re-tested by us ; (†) post-processing corrections or algorithmic bond extraction; (‡) explicit aromatic bonds. Results for methods with † or ‡ are in italics since they cannot be directly compared with purely generative models. An H column ✓ denotes explicit hydrogen generation.

| | | | QM9 | | | | GEOM-Drugs | | |
|---|---|---|---|---|---|---|---|---|---|
| | | | Neutrality | | | | Neutrality | | |
| Model | H | R | Atom(%) | Mol(%) | Val(%) | R | Atom(%) | Mol(%) | Val(%) |
| MolDiff | ✓ | (†, ‡) | - | - | *97.0* | - | - | - | - |
| VoxMol | ✓ | (†) | *99.2* | *89.3* | *98.7* | (†) | *98.1* | *75.0* | *93.4* |
| MiDi | ✓ | (*) | 98.3 | 84.5 | 96.7 | (*,‡) | *96.8* | *31.3* | *62.7* |
| JODO | ✓ | - | 98.5 | 88.2 | 98.1 | (*‡) | *97.0* | *36.7* | *77.8* |
| EDM | ✓ | (*) | 98.1 | 81.9 | 91.6 | (*) | 81.0 | 0.5 | 3.0 |
| GDM-AUG | ✓ | - | 97.6 | 71.6 | 90.4 | - | - | - | |
| EDM-Bridge | ✓ | - | 98.8 | 84.6 | 92.0 | - | - | - | - |
| G-Schnet | ✓ | - | 95.7 | 68.1 | 85.5 | - | - | - | - |
| GeoLDM | ✓ | - | 98.9 | 89.4 | 93.8 | - | 84.4 | - | *99.3* |
| FMG | ✓ | - | **99.4**$_{\pm 0.1}$ | **91.5**$_{\pm 1.4}$ | **98.8**$_{\pm 0.3}$ | - | **94.7** | **16.5** | **64.9** |
| Data | ✓ | - | 99.9 | 99.6 | 99.6 | - | 99.4 | 86.2 | 89.6 |
| MolDiff | - | - | - | - | - | (*, ‡) | - | - | *85.0* |
| FMG | - | - | - | - | - | - | - | - | 80.0 |

Table 2: Molecule graph quality of generated samples from models trained on the QM9 and GEOM datasets, based on 10k samples. We report the mean and standard deviation of the metrics for three different runs (each generating 10k samples) for QM9. See Table 1 for description of symbols.

| | | | QM9 | | | | GEOM-Drugs | | |
|---|---|---|---|---|---|---|---|---|---|
| Model | H | R | MST (↑) | TV$_a$(↓) | TV$_b$(↓) | R | MST (↑) | TV$_a$(↓) | TV$_b$(↓) |
| MiDi | ✓ | (*) | **0.57** | **0.19** | **0.81** | (*,‡) | *0.53* | 1.07 | - |
| EDM | ✓ | (*) | 0.55 | 0.49 | 0.98 | (*) | 0.46 | 1.63 | 5.72 |
| JODO | ✓ | (*) | 0.58 | 0.19 | 0.89 | (*, ‡) | *0.58* | 1.07 | - |
| FMG | ✓ | - | 0.54$_{\pm 0.00}$ | 0.52$_{\pm 0.02}$ | 0.95$_{\pm 0.01}$ | - | **0.49** | 2.4 | **4.78** |
| MolDiff | - | - | - | - | - | (*,‡) | **0.62** | 2.03 | - |
| FMG | - | - | - | - | - | - | 0.50 | **0.63** | 4.58 |

Although our NN architecture is not invariant to SE(3) transformations, our method achieves satisfactory conformational accuracy, with a very low distance between generated and data distributions. We report the $W_1$ distances for bond angles, bond distances, and conformation energy distributions in Appendix A.1, Table 4, and show the corresponding CDFs in Figures 4 and 5.

## 6.2 QM9 CONDITIONAL GENERATION

We employ the same setting as in previous work (Hoogeboom et al., 2022), training molecule property discriminator models on 50% of the QM9 data, and conditional generation models on the remaining 50%. Our generated molecules are then tested using the classifiers. Further details of the experiment and the results are reported in Appendix A.2.

We compare the difference between the highest and lowest occupied molecular energies $\Delta\epsilon$ and the polarizability $\alpha$ of the generated molecules, compared to the true, conditional values. FMG outper-

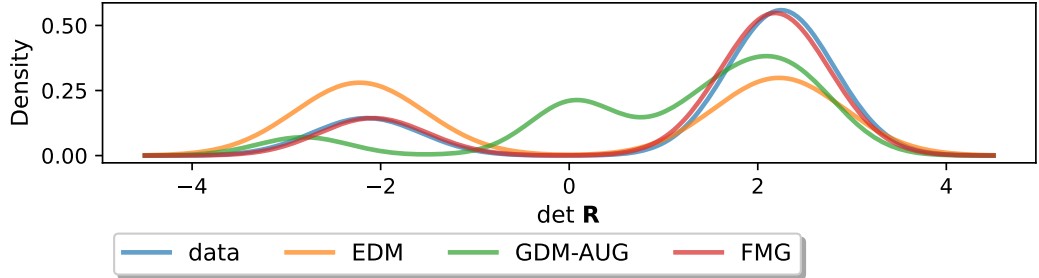

Figure 3: Kernel density estimation of $\det \boldsymbol{R}$, resulted from the data and generated molecules. EDM is unable to distinguish and correctly generate the enantiomer distribution, while the E(3) variant method (GNN) generates improper, 0 volume conformations. FMG matches the enantiomer distribution.

forms EDM in terms of molecular stability, while EDM is better in its fidelity to the conditioning variables $c$.

Additionally, our method generates similar fidelity to the conditioning variable $c$, even without classification-free guidance on the number of atoms $N_a \sim p(N_a|c)$. This could prove useful in situations where the conditional $p(N_a|c)$, cannot be accurately determined based on data statistics (e.g., for generating ligands that interact with specific proteins).

### 6.3 GEOM-DRUGS

The Geometric Ensemble Of Molecules (GEOM) dataset (Axelrod & Gómez-Bombarelli, 2022) contains molecules of up to 181 atoms and 37 million conformations along with their corresponding energies. Following previous work (Hoogeboom et al., 2022; Peng et al., 2023), we select the 30 lowest energy conformers for each molecule graph. Similar to (Peng et al., 2023), we also restrict the data points to only those molecules that contain atoms from $\{$H, C, N, O, F, P, S, Cl$\}$. We extract those molecules that can be contained in the pre-defined evaluated positions that determine the tensors $\mathbf{u}_a$ and $\mathbf{u}_b$, similar to our QM9 experiments. After filtering, the data contains 96.7% of the training molecular conformations and 99.7% unique molecular graph coverage.

We report basic molecule properties in Table 1, where our method achieves highly competitive neutral atoms and molecule percentages. Because of post-generation bond extractions and corrections (marked with (†)) and other bond representation choices, it remains unfortunately difficult to reliably compare against all methods (please refer to Appendix I for details). In Table 2, we see the same effect mentioned in the QM9 experiments, where the method has a worse $TV_a$ metric, likely caused by our method favoring other metrics to correct atom counts. When H atoms are not explicitly generated, our generated atom distribution becomes significantly closer to the data.

The conformational properties of the generated molecules are detailed in Appendix A.1. Our method performs well, even with significantly more complex molecules.

### 6.4 E(3) INVARIANCE AND CHIRALITY ANALYSIS

In this section, we empirically confirm that E(3) invariant implementations lead to invariance wrt enantiomers and show that our proposed method captures all molecular geometries. Our setting consists of a synthetic dataset of 48 chiral molecules with a skewed enantiomer distribution. Details of the experiment are provided in Appendix B.2.

Figure 3 shows the KDEs of generated tetrahedron determinants of $\boldsymbol{R} = [\mathbf{m}_{\text{O}} - \mathbf{m}_{\text{C}}, \mathbf{m}_{\text{N}} - \mathbf{m}_{\text{C}}, \mathbf{m}_{\text{H}} - \mathbf{m}_{\text{C}}]^T \in \mathbb{R}^{3 \times 3}$. The E(3) equivariant generative method (EDM) has $p_\phi(\det \boldsymbol{R}) = p_\phi(-\det \boldsymbol{R})$, which empirically attests Proposition 1.

Table 3: FMG performance with various design choices.

| Model | Val(%) | Neutrality Atom(%) | Neutrality Mol(%) | $TV_a(\downarrow)$ | $TV_b(\downarrow)$ |
|---|---|---|---|---|---|
| EDM | 91.6 | 98.1 | 81.9 | 0.49 | 0.98 |
| $FMG_{aug}$ | 95.8 | 98.6 | 82.6 | 0.69 | 0.94 |
| $FMG_{\beta=0}$ | 98.6 | 99.3 | 92.2 | 1.02 | 1.16 |
| $FMG_{\beta=2}$ | **98.8** | **99.4** | 91.5 | 0.52 | **0.95** |
| $FMG_{\beta=5}$ | 98.6 | **99.4** | 92.2 | **0.48** | 1.02 |
| $FMG_{dist}$ | 97.7 | 99.1 | 90.9 | 0.54 | 1.00 |

We also test a point-cloud method that is variant to reflections. The work of Hoogeboom et al. (2022) (GDM-AUG in Figure 3) provides a variant of their method, where atom positions $\{\mathbf{m}_i\}$ are concatenated to node features. Interestingly, although this method has $p_\phi(\det \boldsymbol{R}) \neq p_\phi(-\det \boldsymbol{R})$, it generates many unnatural conformations, with tetrahedrons of approximately 0 volume.

Finally, FMG captures the distribution well.

## 6.5 ABLATION

We determine the aspects of our method that contribute the most to our competitive performance. In Table 3, $FMG_{aug}$ receives augmented fields based on randomly rotated sets of atom coordinates $\{\mathbf{m}_i\}$ during training, $FMG_\beta$, $\beta \in \{0, 2, 5\}$ varies the degree of classification free guidance on the number of atoms (see Section 4.4), and $FMG_{dist}$ is the performance of the molecular graphs, where bonds are extracted according to bond pairwise atom distances, without considering $\mathbf{u}_b$ channels. The rotation alignment is the main factor that increases the neutrality metric to state-of-the-art performance, as $FMG_{aug}$ performance drops significantly, while classification guidance improves $TV_a$.

For $FMG_{dist}$, we determine bonds based on the lookup table from (Hoogeboom et al., 2022). This confirms that our molecular conformations analyzed in Appendix A.1 are remarkably good since the model's neutrality remains SOTA even when bonds are solely based on the atom positions $\{\mathbf{m}_i\}$.

## 7 CONCLUSION

We demonstrated that continuous field-based molecular representations can achieve highly competitive results. Through theoretical and empirical analysis, we confirmed that E(3) equivariant methods consistently generate enantiomers with equal probabilities. This is crucial, as incorrect specification of enantiomers can lead to a loss of drug effectiveness or severe side effects. We also discuss in detail why developing a corrected SE(3) invariant point-cloud method would be significantly more challenging and show that our alternative successfully generates correct chiral properties.

**Limitations and future work.** While point-cloud methods scale quadratically with the number of atoms, our method depends directly on the size and shape of the molecules. For any new dataset, the dimensions of the field tensors will always dictate the method's computational feasibility and should be carefully considered.

Our tensor dimensions remain fairly large (even with our coarse resolution). Adaptations of current solutions of improved scaling (Gu et al., 2024; Ramesh et al., 2022; Rombach et al., 2022) could, in principle, allow for much larger architectures. In turn, scaling at the expense of group invariances may result in significant strides towards drug-like molecule design (Abramson et al., 2024).

We show significant performance gains using reference rotations, keeping general graph metrics competitive to E(3) formulations while obtaining chiral sensitivity. It remains, however, an open question whether our method can be applied to other 3D object generation tasks.

## 8    ACKNOWLEDGEMENTS

We acknowledge the computational resources provided by the *Aalto Science-IT Project* and *CSC–IT Center for Science, Finland*. This work was supported by *Research Council of Finland* (grants 334600, 359135, 342077), the *Jane and Aatos Erkko Foundation* (grant 7001703), and the *Cancer Foundation of Finland*.

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

# A    ADDITIONAL EXPERIMENTS AND DETAILS

## A.1    CONFORMATIONAL ANALYSIS

We compute the Wasserstein $W_1$ distances for bond angle (BA), and bond lengths (BL) and respectively averaged over atom and bond types. We additionally compute conformational energies ($CE_{MMFF}$ and $CE_{xTB}$) $W_1$ (definitions of these metrics can be found in Appndix H). We report the $W_1$ distances in Tables 4 and 5. In Figures 4 and 5, we also plot the CDF of the distributions, based on which the $W_1$ distances were computed and in Figure 6 and 7 we report the histograms of xTB energies.

Note that for the GEOM-Drugs dataset, MiDi and JODO use explicitly generated aromatic bond types, and thus has a different bond type set $\mathcal{B}$ compared to ours. This also affects the distributions of the bond types 1 and 2 and we therefore omit MiDi from the GEOM-Drugs bond distance metric BL.

Fields cannot enforce the number of atoms $N$ that will be generated, while point clouds do so, sampling $N \sim p_{\mathcal{D}}(N)$ during generation. While not enforcing the number of atoms provides useful flexibility (e.g., in conditional generation, the conditioning may be $p(\cdot|y)$, instead of $p(\cdot|y, N)$, where the latter requires knowledge of the joint $p(y, N)$), we find $TV_a$ (ablation Table 3) and $CE_{xTB}$ (Table 4) to gain significant improvements as a result of atom count conditioning (from no conditioning, $FMG_{\beta=0}$, to $FMG_{\beta=2}$ which was used in this paper), but in this work, we opted for classifier-free guidance on binned atom intervals (see Appendix G), rather than conditioning on a specific atom count. The high impact of atom count conditioning on these metrics provides a strong indication that, if more fine-grained conditioning is adopted, fields will achieve similar $TV_a$ and $CE_{xTB}$ numbers as point-clouds.

Table 4: Conformation results of molecules generated by models trained on the QM9 dataset, based on 10k samples. We report the mean and standard deviation of the metrics for three different runs (each generating 10k samples). The remark column (R) symbols represent: (*) re-tested by us. Reported bond length W1 distances (BL) are multiplied by 100.

| Model | R | BL ($\downarrow W_1$) | BA ($\downarrow W_1$) | $CE_{MMFF}$ ($\downarrow W_1$) | $CE_{xTB}$ ($\downarrow W_1$) |
|---|---|---|---|---|---|
| EDM | (*) | **0.47** | **0.68** | 3.45 | 0.30 |
| MiDi | (*) | 1.92 | 0.95 | 3.12 | 0.34 |
| JODO | (*) | 0.62 | 0.83 | **1.41** | **0.22** |
| FMG | - | $0.59_{\pm 0.01}$ | $1.08_{\pm 0.12}$ | $2.15_{\pm 0.385}$ | $0.71_{\pm 0.02}$ |
| $FMG_{\beta=0}$ | - | - | - | - | 0.98 |

Table 5: Conformation results of molecules generated by models trained on the GEOM-Drugs, based on 10k samples. The remark column (R) symbols represent: (*) re-tested by us ; (†) post-processing corrections or algorithmic bond extraction; (‡) explicit aromatic bonds. Reported bond length W1 distances (BL) are multiplied by $10^2$.

| Model | R | BL ($\downarrow W_1$) | BA ($\downarrow W_1$) | $CE_{MMFF}$ ($\downarrow W_1$) | $CE_{xTB}$ ($\downarrow W_1$) |
|---|---|---|---|---|---|
| EDM | (*) | 8.84 | 4.36 | - | 1.93 |
| MiDi | (*,‡) | - | 1.38 | 40.98 | 1.82 |
| JODO | (*,‡) | - | **0.351** | **7.59** | **1.21** |
| FMG | - | **2.74** | 1.00 | 53.14 | 3.22 |

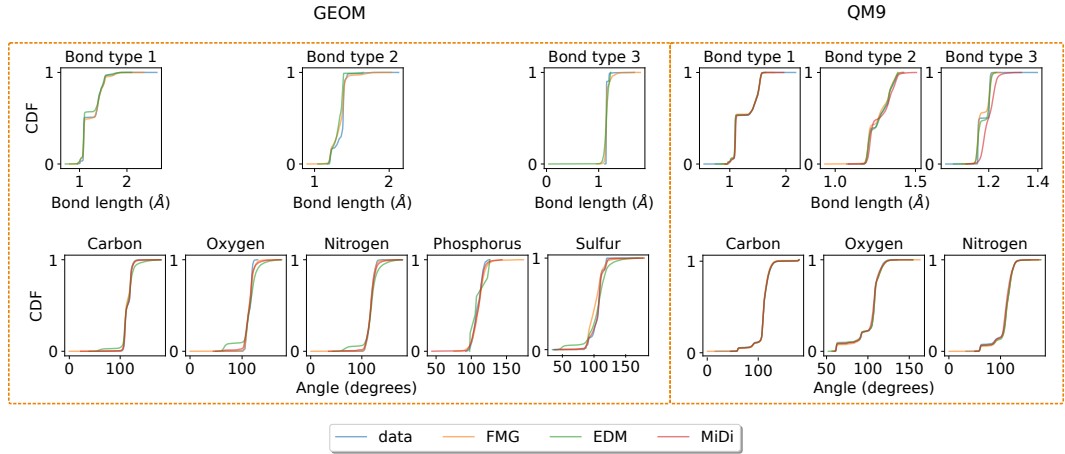

Figure 4: Cumulative distribution function plot of generated and training data for MiDi, EDM, and FMG for the GEOM-Drugs and QM9 datasets. All models capture these conformational properties very well on QM9, and reasonably well on GEOM.

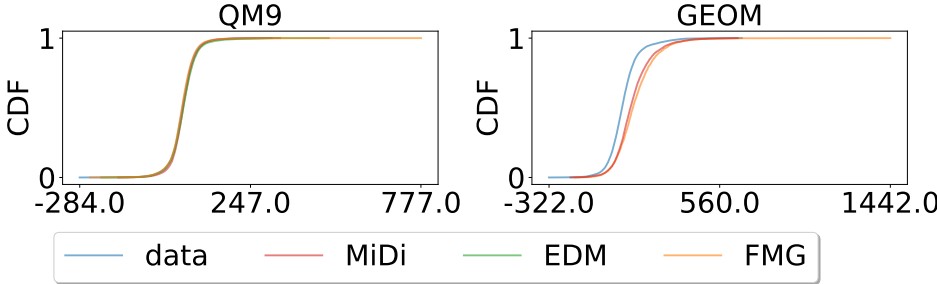

Figure 5: Cumulative distribution function plot of generated and training data energy conformations (kcal/mol) for MiDi, EDM, and FMG for the GEOM-Drugs and QM9 datasets. On GEOM, not enough valid samples were obtained to compute the energy distribution. Both models generate higher energies than the data on GEOM-Drugs (notably, MiDi was trained on a subset of lower energy molecules on GEOM).

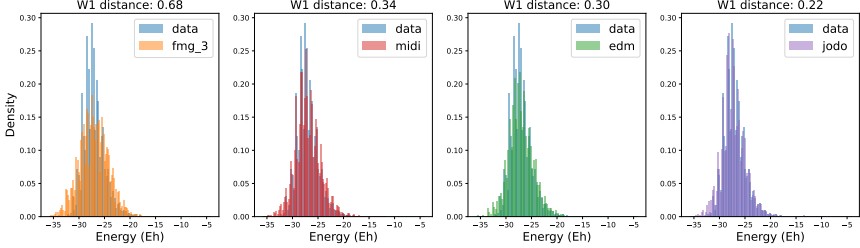

Figure 6: xTB energy conformations (Eh) for MiDi, EDM, JODO, and FMG for the QM9 dataset.

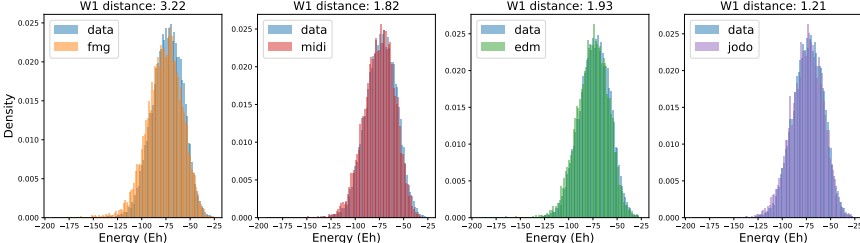

Figure 7: xTB energy conformations (Eh) for MiDi, EDM, JODO, and FMG for the GEOM-Drugs dataset.

### A.2 CONDITIONAL GENERATION

We train classifiers $\phi_c$ for the two considered properties, $\alpha$ and $\Delta\epsilon$. The property classifiers $\phi_c$ are E(3) invariant graph neural networks that receive pairwise distances between atoms $||\mathbf{m}_i - \mathbf{m}_j||$ and atom types as categorical features, similar to the architecture used for parametrizing EDM. We follow the setting from (Hoogeboom et al., 2022), splitting the 100 thousand molecules into two equal subsets, and training our generative method and property classifiers on each of the two subsets, respectively.

To sample the properties $c \sim p_{\mathcal{D}}$, we split molecular properties in one thousand equal bins, and form, based on training data statistics, the conditional and marginal categorical distributions $p(N|\mathbf{c}_i)$, and $p(\mathbf{c}_i)$, where $\mathbf{c}_i$ is the $i^{\text{th}}$ property bin and $N$ is the number of atoms. We test our conditional model by sampling two thousand pairs of $[\mathbf{c}_i, N] \sim p(\mathbf{c}_i)p(N|\mathbf{c}_i)$, and further sample the actual conditional property $c \sim \mathcal{U}(\mathbf{c}_{i,0}, \mathbf{c}_{i,1})$, with which we condition the generative model on $\epsilon_\theta(\mathbf{f}_t, t, N_a, c)$.

We test our generative models by predicting the generated molecules' properties, and computing the $L_1$ loss between those predictions and the conditioning variable $c$ used during generation.

In Table 6, FMG is the trained conditional method for the two properties, respectively, and FMG$_{\beta=0}$ is the same model, where we do not use classification-free guidances on the number of atoms according to the property $p(N|c)$.

We also include additional baselines: FMG$_{\text{uc}}$ is an unconditional model, where we sample molecules according to an appropriate number of atoms $p(N_a|c)$, and FMG$_{\text{rnd}}$ are completely random samples, from an unconditional model.

Table 6: Conditional generation results. Atom and Mol Neutrality are reported in percentages. The mean results of 2k samples generated by our methods are shown.

| Model | $\alpha$ | | | $\Delta\epsilon$ | | | |
| | | Neutrality | | | Neutrality | | |
| | L1($\downarrow$) | Atom | Mol | L1 ($\downarrow$) | Atom | Mol | Conditional |
|---|---|---|---|---|---|---|---|
| EDM | **2.76** | - | 80.4 | **655** | - | 81.7 | ✓ |
| FMG | 3.55 | 99.0 | **86.8** | 746 | 98.9 | **84.5** | ✓ |
| FMG$_{\beta=0}$ | 3.92 | 99.0 | 86.7 | 734 | 98.6 | 83.0 | ✓ |
| FMG$_{\text{uc}}$ | 6.92 | 99.4 | 91.5 | 1259 | 99.4 | 91.5 | |
| FMG$_{\text{rnd}}$ | 9.03 | 99.4 | 91.5 | 1451 | 99.4 | 91.5 | |

We implement property conditioning by shift-scaling the channels, as presented in (Perez et al., 2018), for all U-Net layers (similar to time and number of atoms conditioning). Note that we did not use classifier-free guidance for molecular properties $c$.

# B  CHIRALITY ANALYSIS AND E(3) INVARIANCE

In Appendix B.1, we develop our theoretical analysis and in Appendix B.2 we describe our chirality experiment details.

## B.1  THEORETICAL ANALYSIS

In Appendix B.1.1, we draw the connection between a NN's E(3) invariance and its resulting invariance to enantiomer pairs.

In Appendix B.1.2, we show how point-cloud Diffusion models parametrized by E(3) invariant NN become E(3) equivariant.

We give proofs of Proposition1, Lemma 2, and Proposition 3, in Appendix B.1.3, B.1.4, and B.1.5. In Appendix B.1.6 we discuss cases where Proposition 3 applies.

### B.1.1  E(3) INVARIANCE OF NNS AND CHIRALITY

Let $\mathbf{v}_1, \mathbf{v}_2 \in \mathbb{R}^3$ be two, three-dimensional vectors, and let $\mathbf{d} = \mathbf{v}_1 - \mathbf{v}_2$. By definition, orthogonal matrices $\boldsymbol{Q}$ have an inverse equal to $\boldsymbol{Q}^{-1} = \boldsymbol{Q}^T$. Therefore, it is easy to see that $||\boldsymbol{Q}\mathbf{d}|| = ||\mathbf{d}||$ since

$$||\boldsymbol{Q}\mathbf{d}||^2 = (\boldsymbol{Q}\mathbf{d})^T(\boldsymbol{Q}\mathbf{d}) = \mathbf{d}^T\boldsymbol{Q}^T\boldsymbol{Q}\mathbf{d} = \mathbf{d}^T\boldsymbol{I}\mathbf{d} = \mathbf{d}^T\mathbf{d} = ||\mathbf{d}||^2, \tag{15}$$

where $\boldsymbol{I}$ is the identity matrix, and $|| \cdot ||$ is the $L_2$ norm. Since reflection and rotation matrices are orthogonal, it follows, that the Euclidean distances between any two vectors $\mathbf{v}_1, \mathbf{v}_2$ are invariant to orthogonal matrix multiplication, and, therefore, to any rotation or reflection matrix.

Consider now the cosine of the angle between any two vectors $\mathbf{v}_1, \mathbf{v}_2 \in \mathbb{R}^3$, transformed by an orthogonal matrix $Q$. We have that

$$\cos(\boldsymbol{Q}\mathbf{v}_1, \boldsymbol{Q}\mathbf{v}_2) = \frac{\boldsymbol{Q}\mathbf{v}_1 \cdot \boldsymbol{Q}\mathbf{v}_2}{||\boldsymbol{Q}\mathbf{v}_1||||\boldsymbol{Q}\mathbf{v}_2||} = \frac{(\boldsymbol{Q}\mathbf{v}_1)^T(\boldsymbol{Q}\mathbf{v}_2)}{||\mathbf{v}_1||||\mathbf{v}_2||} = \frac{\mathbf{v}_1^T\boldsymbol{I}\mathbf{v}_2}{||\mathbf{v}_1||||\mathbf{v}_2||} = \frac{\mathbf{v}_1 \cdot \mathbf{v}_2}{||\mathbf{v}_1||||\mathbf{v}_2||} = \cos(\mathbf{v}_1, \mathbf{v}_2), \tag{16}$$

where we have used equation 15 for $||\boldsymbol{Q}\mathbf{v}_i|| = ||\mathbf{v}_i||$. All distances and angles between all vectors in the $\mathbb{R}^3$ space are therefore preserved.

Any function $\phi$ (e.g., a neural network) that receives pairwise atom distances or angles will therefore be invariant to orthogonal matrix multiplication and implicitly, to reflections.

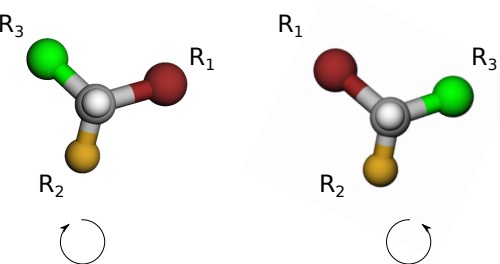

Figure 8: R and S configurations of a general tetrahedral configuration with four, distinct residues (the 4th atom, perpendicular to the planar perspective was not annotated for simplicity). The determinant $\det[\mathbf{m}_{R1} - \mathbf{m}_C, \mathbf{m}_{R2} - \mathbf{m}_C, \mathbf{m}_{R3} - \mathbf{m}_C]$ is constant under rotations, but changes sign after reflections. Equivalently, the sign changes when swapping $\mathbf{m}_{R1}$ and $\mathbf{m}_{R3}$ positions, which defines the clockwise (left) or counterclockwise (right) orientation. The determinant sign or, equivalently, the orientation determines the enantiomer R or S configuration.

Consider a tetrahedral configuration, where the central carbon atom is positioned at $\mathbf{m}_C$, and is connected to four unique residues $R1, R2, R3$, and $R4$, initially positioned at $\mathbf{m}_{R1}, \mathbf{m}_{R2}, \mathbf{m}_{R3}$, and $\mathbf{m}_{R4}$, respectively (see Figure 8). Observing the molecule along the $R_4$-C-axis, a way of defining the chirality of the molecule is to specify the ordering of the remaining residues in clockwise or counterclockwise orientation. Assuming we start with $R2$ position fixed at $\mathbf{m}_{R2}$, swapping $R1$ and $R3$ positions has the effect of reversing the rotation direction and creating a distinct chiral configuration. Now, considering the plane defined by $\mathbf{m}_{R2}$, the central carbon position $\mathbf{m}_C$, and $\mathbf{m}_{R4}$, the same effect can be achieved by a reflection about this plane.

By Householder transformation, a reflection about a hyperplane (in this case, plane), can be defined by the plane's normal vector (perpendicular to the plane and of norm 1), for any plane that passes the origin:

$$\boldsymbol{P} = \boldsymbol{I} - 2\mathbf{v}\mathbf{v}^T, \tag{17}$$

where the reflection matrix $\boldsymbol{P}$ leaves any vector on the plane unchanged. Translating the object by, e.g., $-(\mathbf{m}_{R2} + \mathbf{m}_C + \mathbf{m}_{R4})/3$, the plane now passes the origin, meaning we can determine such a matrix $\boldsymbol{P}$ for the $\mathbf{m}_{R2}, \mathbf{m}_C, \mathbf{m}_{R4}$, and swap $\mathbf{m}_{R_3}$ and $\mathbf{m}_{R_1}$ with that reflection.

In general, a reflection about any plane can be achieved by a specific translation, a rotation, and a reflection, applied to the whole $\mathbb{R}^3$ space (after reflection, the molecule may be moved back with an opposite translation and rotation). As Euclidean distances and angles between any two vectors are preserved under each of these operations, it follows that Euclidean distances and angles become invariant to the chirality.

To our knowledge, all current point-cloud-based methods use Euclidean distances or relative angles between atom position vectors $\mathbf{m}_i, \mathbf{m}_j$, e.g., for message passing in their GNN architectures. These methods cannot distinguish enantiomers. As such, their resulting E(3) invariant probability distributions will always produce them completely at random, and the molecular graphs will not change based on whether the S or R enantiomer configuration is present in that molecule.

### B.1.2 E(3) INVARIANCE OF THE GENERATIVE SETTING (DDPM)

The problem arising from E(3) invariant neural networks in molecule generation is the E(3) equivariance of the resulting conditional probability distribution $p_\phi(\mathbf{m}_s|\mathbf{m}_t)$. Concretely, for a set of positions $\mathbf{m} = \{\mathbf{m}_1, \dots, \mathbf{m}_N\}, \mathbf{m}_i \in \mathbb{R}^3$, the R and S enantiomer configurations of molecules will be equally likely, since the marginal becomes E(3) invariant $p_\phi(\mathbf{m}) = p_\phi(R\mathbf{m}), \forall R$ orthogonal. We illustrate this in the context of EDM (Hoogeboom et al., 2022). Consider a set of $\mathbb{R}^3$ points $\{\mathbf{m}_1, \dots, \mathbf{m}_N\}$ of a molecule containing $N$ atoms and a neural network $\phi_\mathbf{m}$. At each layer $l$ in the GNN network parametrizing EDM, positions are updated as follows:

$$\mathbf{m}_i^{l+1} = \mathbf{m}_i^l + \sum_{i \neq j} \frac{\mathbf{m}_i^l - \mathbf{m}_j^l}{d_{ij} + 1} \phi_\mathbf{m}(d_{ij}, \dots), \tag{18}$$

where $\phi_\mathbf{m}$ receives atom pair Euclidean distances and other features that are E(3) invariant.

Every layer is a linear combination of atom positions $\mathbf{m}_i$ with E(3) invariant coefficients that are equal to 1 or have the form $\{\pm\phi_\mathbf{m}(d_{ij}, \dots)/(1 + d_{ij})\}$. The update at layer $l + 1$ of atom $i$ is then given by $\mathbf{m}_i^{l+1} = T_i^{l+1}(\mathbf{m}^l)$, and since $T_i^{l+1}$'s coefficients are E(3) invariant, the equality $T_i^{l+1}(\boldsymbol{R}\mathbf{m}^l) = \boldsymbol{R}T_i^{l+1}(\mathbf{m}^l)$ holds for all orthogonal matrices $\boldsymbol{R}$.

Consider now the $\epsilon_\phi$ parametrization of the reverse DDPM process

$$p_\phi(\mathbf{m}_s|\mathbf{m}_t, t) = \mathcal{N}(\mathbf{m}_s; \boldsymbol{\mu}_\phi(\mathbf{m}_s, s, t), \boldsymbol{\Sigma}(s, t)), \tag{19}$$

where

$$\boldsymbol{\mu}_\phi(\mathbf{m}_t; s, t) = \frac{1}{\alpha_{ts}}\mathbf{m}_t - \frac{\sigma_{ts}^2}{\alpha_{ts}\sigma_t}\epsilon_\phi(\mathbf{m}_t, t), \tag{20}$$

where $\alpha_{ts} = \alpha_t/\alpha_s$ and $\sigma_{ts}^2 = \sigma_t^2 - \alpha_{ts}^2\sigma_s^2$, keeping the notation from Section 4.2. It follows that

$$\begin{aligned}
\boldsymbol{\mu}_\phi(\boldsymbol{R}\mathbf{m}_t; s, t) =& \frac{1}{\alpha_{ts}}\boldsymbol{R}\mathbf{m}_t - \frac{\sigma_{ts}^2}{\alpha_{ts}\sigma_t}\epsilon_\phi(\boldsymbol{R}\mathbf{m}_t, t) \\
=& \frac{1}{\alpha_{ts}}\boldsymbol{R}\mathbf{m}_t - \frac{\sigma_{ts}^2}{\alpha_{ts}\sigma_t}\boldsymbol{R}\epsilon_\phi(\mathbf{m}_t, t) & \text{equivariance of } \epsilon_\phi \\
=& \boldsymbol{R}(\frac{1}{\alpha_{ts}}\mathbf{m}_t - \frac{\sigma_{ts}^2}{\alpha_{ts}\sigma_t}\epsilon_\phi(\mathbf{m}_t, t)) \\
=& \boldsymbol{R}\boldsymbol{\mu}_\phi(\mathbf{m}_t; s, t),
\end{aligned}$$

Since the covariance matrix of the reverse process $\boldsymbol{\Sigma}$ does not depend on positions $\mathbf{m}$, and because of the $\boldsymbol{\mu}_\phi$ equivariance wrt E(3), we have that

$$p_\phi(\mathbf{m}_s|\mathbf{m}_t, t) = p_\phi(\boldsymbol{R}\mathbf{m}_s|\boldsymbol{R}\mathbf{m}_t, t), \forall t. \tag{21}$$

We give a short version of the proof from (Xu et al., 2022; Hoogeboom et al., 2022), which shows how the conditional $p_\phi(\mathbf{m}_s|\mathbf{m}_t, t)$ and marginal $p_\phi(\mathbf{m}_1)$ being, respectively, equivariant and invariant to a symmetry group, results in the invariance of $p_\phi(\mathbf{m}_0)$. Following (Hoogeboom et al., 2022), we have

$$\begin{aligned}
p(\boldsymbol{R}\mathbf{m}_s) =& \int_{\mathbf{m}_t} p(\boldsymbol{R}\mathbf{m}_s|\mathbf{m}_t)p(\mathbf{m}_t) \\
=& \int_{\mathbf{m}_t} p(\boldsymbol{R}\mathbf{m}_s|\boldsymbol{R}\boldsymbol{R}^{-1}\mathbf{m}_t)p(\boldsymbol{R}\boldsymbol{R}^{-1}\mathbf{m}_t) \\
=& \int_{\mathbf{m}_t} p(\mathbf{m}_s|\boldsymbol{R}^{-1}\mathbf{m}_t)p(\boldsymbol{R}^{-1}\mathbf{m}_t) & \text{Invariance of } p(\mathbf{m}_t) \text{ \& equivariance of } p(\mathbf{m}_s|\mathbf{m}_t) \\
=& \int_{\mathbf{m}'} p(\mathbf{m}_s|\mathbf{m}')p(\mathbf{m}') \cdot |\det \boldsymbol{R}| & \text{Change of variables } \mathbf{m}' = \boldsymbol{R}\mathbf{m}_t \\
=& p(\mathbf{m}_s),
\end{aligned}$$

where we note that the change of variables $\mathbf{m}' = \boldsymbol{R}\mathbf{m}_t$ also changes the integration volume, and the equality stands for both $\det \boldsymbol{R} = \pm 1$ (rotation and reflection matrices).

Observing that $p(\mathbf{m}_1) = \mathcal{N}(\mathbf{0}, \mathbf{I})$ is invariant to orthogonal matrix multiplication, it follows that $p(\mathbf{m}_0) = p(\boldsymbol{R}\mathbf{m}_0)$ for all orthogonal matrices $\boldsymbol{R}$ (induction). We have shown how reflection matrices $\boldsymbol{R}$ switch the R and S chiral configurations of a molecule at the beginning of this section. This result can be empirically observed in the orange line of the main text Figure 3.

### B.1.3 PROPOSITION 1 PROOF.

Consider atom positions $\mathbf{m} \in \mathbb{R}^{3\times N}$ and $\mathbf{m}' \in \mathbb{R}^{3\times N}$ from enantiomer R and S configurations, and $p_\phi$ an E(3) invariant probability distribution. E(3) invariance implies SE(3) invariance, since SE(3)$\subset$E(3). This gives

$$p_\phi(\mathbf{m}) = p_\phi(\boldsymbol{R}_+\mathbf{m} + \mathbf{t}), \tag{22}$$

for all rotation matrices $\boldsymbol{R}_+$ (orthogonal, with $\det \boldsymbol{R}_+ = 1$) and $\mathbf{t}$ is a translation vector. The E(3) invariance of $p_\phi$ gives

$$p_\phi(\boldsymbol{R}_-(\boldsymbol{R}_+\mathbf{m} + \mathbf{t})) = p_\phi(\boldsymbol{R}_+\mathbf{m} + \mathbf{t}) = p_\phi(\mathbf{m}), \tag{23}$$

where $\boldsymbol{R}_-$ is orthogonal, with $\det \boldsymbol{R}_- = -1$.

We restate that swapping two residue positions $\mathbf{m}_{R1}$ and $\mathbf{m}_{R3}$ of atoms $R1$ and $R3$ connected to a chiral center will change the clockwise or counterclockwise orientation (see Figure 8). Swapping

$\mathbf{m}_{R1}$ and $\mathbf{m}_{R3}$ positions can be achieved through a reflection about the plane defined by the points $\{\mathbf{m}_{R4}, \mathbf{m}_C, \mathbf{m}_{R2}\}$. In turn, a reflection about a plane can be expressed as a rotation, a translation, and a reflection. This implies that $\exists \boldsymbol{R}_+, \boldsymbol{R}_-, \mathbf{t}$, s.t. $\mathbf{m}' = \boldsymbol{R}_-(\boldsymbol{R}_+\mathbf{m} + \mathbf{t})$, which gives

$$p_\phi(\mathbf{m}') = p_\phi(\boldsymbol{R}_-(\boldsymbol{R}_+\mathbf{m} + \mathbf{t})) = p_\phi(\mathbf{m}). \tag{24}$$

### B.1.4 LEMMA 2 PROOF.

Let $M' = \{\mathbf{m}_1, \ldots, \mathbf{m}_m\}$, $m \leq n, \mathbf{m}_i \in \mathbb{R}^n$ be $n$-dimensional vectors, and $f : \mathbb{R}^{m \times n} \to S$ be an SE(n) invariant function. Center the object in 0, creating

$$M = \{\mathbf{m}_1 - \bar{\mathbf{m}}, \ldots, \mathbf{m}_m - \bar{\mathbf{m}}\} \tag{25}$$

where $\bar{\mathbf{m}} = \frac{1}{m} \sum_{i=1}^m \mathbf{m}_i$. Observe how the vectors of $M$ are linearly dependent, since

$$\sum_{i=1}^m (\mathbf{m}_i - \bar{\mathbf{m}}) = \sum_{i=1}^m (\mathbf{m}_i) - n \cdot \frac{1}{n} \sum_{i=1}^m \mathbf{m}_i = \mathbf{0} \tag{26}$$

This means that $dim(M) \leq m - 1 \leq n - 1$, which gives $dim(M^\perp) \geq 1$, since $M$ contains at most $n - 1$ linearly independent vectors. This gives

$$M^\perp = \{\mathbf{x} \in \mathbb{R}^n | \langle \mathbf{x}, \mathbf{m} \rangle = 0, \forall \mathbf{m} \in Span(M)\} \neq \{\mathbf{0}\}. \tag{27}$$

Let $\mathbf{p} \in M^\perp$ be any of $M$'s hyperplane normal vectors (perpendicular to all vectors in $M$ and unit length). Using the Householder transformation, define a reflection matrix about the hyperplane using

$$\boldsymbol{P} = \boldsymbol{I} - 2\mathbf{p}\mathbf{p}^T. \tag{28}$$

Note how any vector in $M$ is unchanged when reflecting by $\mathbf{P}$, since

$$\boldsymbol{P}\mathbf{m} = (\boldsymbol{I} - 2\mathbf{p}\mathbf{p}^T)\mathbf{m} = \mathbf{m} - 2\mathbf{p}\mathbf{p}^T\mathbf{m} = \mathbf{m} - \mathbf{0}, \forall \mathbf{m} \in M. \tag{29}$$

For all $\boldsymbol{R}$, $det(\boldsymbol{R}) = -1$ and $\boldsymbol{R}$ orthogonal (reflection matrix), note how

$$f(M) = f(\boldsymbol{R}\boldsymbol{P}M) \quad \text{since } \boldsymbol{R}\boldsymbol{P} \text{ is a proper rotation matrix and f is SE(3) invariant} \tag{30}$$
$$= f(\boldsymbol{R}M) \quad \text{since } \boldsymbol{P}M = M \tag{31}$$

where $\boldsymbol{R}\boldsymbol{P}$ is a proper rotation matrix since it is orthogonal and $det(\boldsymbol{R}\boldsymbol{P}) = det(\boldsymbol{R})det(\boldsymbol{P}) = -1 \cdot -1 = 1$.

We have established that $f$ is reflection invariant for 0-centered objects of $m \leq n$ $\mathbb{R}^n$ vectors. We will utilize the translation invariance of $f$ to extend the proof to any $M'$ object of $m \leq n$ $\mathbb{R}^n$ vectors.

Let $T_\mathbf{v}$ be a translation operator applied to a set of vectors:

$$T_\mathbf{v}(M') = \{\mathbf{m}_1 + \mathbf{v}, \ldots, \mathbf{m}_m + \mathbf{v}\}. \tag{32}$$

Let $\boldsymbol{R}$ be a reflection matrix. Then

$$\begin{aligned}
f(\boldsymbol{R}M') &= f(\boldsymbol{R}T_{\bar{\mathbf{m}}}(M)) && \text{from eq. 25, } M' = T_{\bar{\mathbf{m}}}(M) \\
&= f(\boldsymbol{R}\{\mathbf{m}_1 - \bar{\mathbf{m}} + \bar{\mathbf{m}}, \ldots, \mathbf{m}_m - \bar{\mathbf{m}} + \bar{\mathbf{m}}\}) \\
&= f(\{\boldsymbol{R}\mathbf{m}_1 - \boldsymbol{R}\bar{\mathbf{m}} + \boldsymbol{R}\bar{\mathbf{m}}, \ldots, \boldsymbol{R}\mathbf{m}_m - \boldsymbol{R}\bar{\mathbf{m}} + \boldsymbol{R}\bar{\mathbf{m}}\}) \\
&= f(T_{\boldsymbol{R}\bar{\mathbf{m}}}(\boldsymbol{R}M)) \\
&= f(\boldsymbol{R}M) && \text{translation invariance} \\
&= f(M) && \text{equation 31} \\
&= f(T_{\bar{\mathbf{m}}}(M)) && \text{translation invariance} \\
&= f(M'),
\end{aligned}$$

which means that for any set of $m \leq n$ $n$-dimensional vectors $M'$, an SE(n) invariant function $f$ has $f(\boldsymbol{R}M') = f(M')$. Together with translation and rotation invariances of $f$, this implies that $f$ is E(n) invariant.

**Remark 1.** *If $f : \mathbb{R}^{k \times n} \to S, k \leq n$ is SE(n) invariant (equivalently E(n) invariant), and $g : S^m \to P$, then $g(f(M_1), \ldots f(M_m))$, with $M_i \in \mathbb{R}^{k \times n}$ is E(n) invariant.*

The remark is made by applying any translation, rotation, and reflections to $M_i$ which leaves $f$ unchanged. And will be useful for our next proof.

### B.1.5 PROPOSITION 3 PROOF.

**Proposition 3 proof.** Assume $n$ vectors $\mathbf{m}_1, \ldots, \mathbf{m}_n$ in $\mathbb{R}^3$, where $n \geq 5$. Also, assume $f : \mathbb{R}^{m \times 3} \to S$ is an SE(3) invariant function, but not E(3) invariant. We know $m \geq 4$, by Lemma 2. Consider $f(\mathbf{m}_1, \mathbf{m}_2, \mathbf{m}_3, \mathbf{m}_4) = \text{sign}(det([\mathbf{m}_1 - \mathbf{m}_4, \mathbf{m}_2 - \mathbf{m}_4, \mathbf{m}_3 - \mathbf{m}_4]))$, and note that it is SE(3) invariant (rotations and translations leave the function unchanged, whereas reflections change its sign).

**Remark 2.** *Enantiomer configuration is a local property, which can only be determined by at least 4 vectors from its set of 5 atoms (4 residues and one central atom), and $f$ can be used to determine chirality. Throughout this proof, we refer to "chirality" as the clockwise or counterclockwise atom arrangements when looking down from one vertex of a tetrahedron.*

Let $M$ and $M'$ be 4 atom subsets. Assume any triplet subset of $M$ has three linearly independent vectors. Then, $f(M)$ conveys no information on any other set $f(M')$, unless $M = M'$. Intuitively, choose any plane from $M$. The resulting determinant's sign, which distinguishes chiral centers, will depend on what side of the plane we choose the 4th point. The fact that $f$ can be used to determine chirality stems from the equivalence between the classical chirality definition (using rotations) and determinant signs (see the beginning of Appendix B.1). All 4 sets of atoms need to be evaluated to have all potential enantiomer states that can happen in $n$ atom molecules.

We now observe that we do not strictly need to compute $f(\cdot)$ for all ordered sets of 4 atoms. For a set of 4 atoms, observe how their permutations may, at most, the sign:

$$f(\mathbf{m}_1, \mathbf{m}_2, \mathbf{m}_3, \mathbf{m}_4) = f(\pi_e(\mathbf{m}_1, \mathbf{m}_2, \mathbf{m}_3, \mathbf{m}_4)) = -f(\pi_o(\mathbf{m}_1, \mathbf{m}_2, \mathbf{m}_3, \mathbf{m}_4)), \tag{33}$$

where we denote by $\pi_e$ and $\pi_o$ even and odd permutation, respectively. Therefore, we need $\binom{n}{4}$ to compute all unordered sets of 4 atoms, giving $\mathcal{O}(n^4)$.

**Remark 3.** *The $\binom{n}{4}$ features $f(\mathbf{m}_{i1}, \mathbf{m}_{i2}, \mathbf{m}_{i3}, \mathbf{m}_{i4})$ need paired atom type information $(\mathbf{a}_{i1}, \mathbf{a}_{i2}, \mathbf{a}_{i3}, \mathbf{a}_{i4})$ using a function that is order-variant on atoms, meaning.*

$$g(f(\mathbf{m}_{i1}, \mathbf{m}_{i2}, \mathbf{m}_{i3}, \mathbf{m}_{i4}), (\mathbf{a}_{i1}, \mathbf{a}_{i2}, \mathbf{a}_{i3}, \mathbf{a}_{i4})) \neq g(f(\mathbf{m}_{i1}, \mathbf{m}_{i2}, \mathbf{m}_{i3}, \mathbf{m}_{i4}), (\pi(\mathbf{a}_{i1}, \mathbf{a}_{i2}, \mathbf{a}_{i3}, \mathbf{a}_{i4}))), \tag{34}$$

where $\pi$ is any permutation except the identity.

### B.1.6 CASES WHERE PROPOSITION 3 APPLIES

Consider the distributions $p_1, \ldots, p_n$, with $p_i(\mathbf{m}_i) = \mathcal{N}(\mathbf{0}, \mathbf{I}), \forall i$ and random associated atom types $\{a_i\}$ (also drawn from an uninformative distribution). The task is transforming these distributions into the conformer distribution of $n$ atom molecules, some of which contain chiral centers. By the uninformative assumption, we need to compute the chirality of all sets of 5 atoms, since all sets have equal probability of being chiral. By remarks 1, 2 and 3, this scales prohibitively, in $\mathcal{O}(n^4)$.

Chiral-aware, SE(3) invariance becomes intractable in point-cloud, diffusion models since the occurrence of reflections needs to be assessed in local neighborhoods (we are specifically interested in potential reflections of chiral centers, but all atoms may be part of chiral centers equally likely). For diffusion methods that define uninformative and identically distributed vector positions at $t = 1$, it implies that all sets of 4 atoms may be part of tetrahedrons at $t = 0$.

Examples of E(3) invariant diffusion models operating on point-clouds are (Hoogeboom et al., 2022; Peng et al., 2023; Vignac et al., 2023b; Xu et al., 2023; Wu et al., 2022). All these methods implement NN $\phi$ activating on sets of $\{f(M_1), \ldots, f(M_n)\}$, where $M_1 \in \mathbb{R}^{2 \times 3}$, and $f$ are Euclidean norms $f(M_i) = ||M_{i,1} - M_{i,2}||_2$. They are E(3) invariant by Remark 1, and can only be made chiral-aware using $\mathcal{O}(n^4)$ scaling, by Proposition 3.

### B.2 EXPERIMENT DETAILS

We create a dataset of 48 molecules, each one having a central tetrahedral carbon that is always connected to the set of four atoms $\mathcal{C} = \{C, H, N, O\}$. We create multiple unique molecules, by connecting the three atoms $C, N, O \in \mathcal{C}$ to other arbitrary residues. In Figure 9, we show half of the molecular graphs within our data. For each of them, we create molecular conformations using RDKit. Note how in this experiment we are not necessarily interested in creating realistic molecules, but simply any molecules that contain chiral configurations.

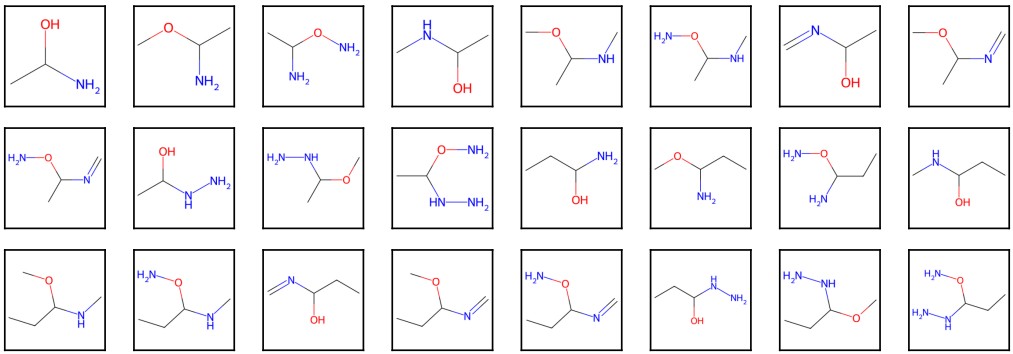

Figure 9: Molecular graphs of half of the chiral dataset.

We train FMG for 10k iterations, and EDM for 300k iterations, and test it for multiple checkpoints. Since the dataset is small, we do not use mini-batches and set our batch size to the data length.

## C FIELD PARAMETERS

For the QM9 dataset, we determined that a resolution of 0.33Å and $32^3$ voxels for each molecule is a good compromise between the sample size and the percentage of the dataset's samples (the total size being $\mathbf{u} \in \mathbb{R}^{8 \times 32 \times 32 \times 32}$, for 5 and 3 atom and bond channels, respectively). We set the RBF component's $\sigma = 0.224$, for all fields, and $\gamma = 0.176$. The component values $\gamma = 0.176$ were determined as the maximum value of an RBF component having its positions $\mathbf{m}$ in one of the evaluated positions $\mathcal{G}$. Although the field parameters do not reflect the actual, physical size of the atoms, we note how true physical fields may be recovered by appropriately scaling the variance of each atom RBF component, based on the channels $a$ they belong to.

We used the same resolution for GEOM. After rotating all molecules using the principle components of the atom position vectors, the resulting tensors $\mathbf{u} \in \mathbb{R}^{11 \times 64 \times 40 \times 32}$ were enough to cover 96.66% of the total data points and 99.71% unique molecular graphs.

The field hyperparameters were selected during early experimentation. We focused on the effect of field hyperparameters on fitting $u_a(x)$ and $u_b(x)$ to the generated fields $\boldsymbol{u}_a$ and $\boldsymbol{u}_b$ (Equations 13 and 14 and Appendix E). We compute the "extraction accuracy", which measures how many molecules have the same molecular graph extracted from the fields they create. We additionally report the average "distance from original", computed as the average Euclidean distance between the correct atom positions and the ones extracted. For both, we determine how robust they are for various levels of noise applied to the fields. Ablation of field hyperparameters for the resulting diffusion model's accuracy was not performed, due to the heavy computational resources required.

**Field standard deviation** $\sigma$: Intuitively, larger standard deviations $\sigma$ would facilitate smaller receptive field CNN kernels to capture longer distance neighboring RBF components. At the same time, we require that molecular graphs and conformations are correctly extracted. Figure 10 shows how increasing $\sigma$ worsens the extraction accuracy, and our choice ($\sigma^2 = 0.05$) prioritized molecular conformation (distance from original) and graph (Extraction accuracy) reliability over noise levels.

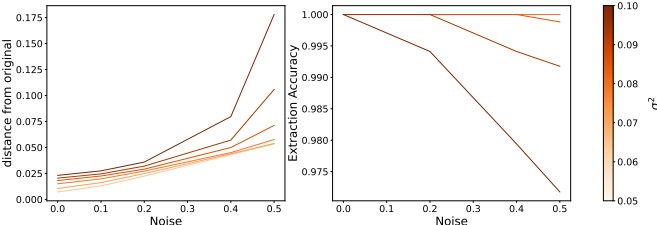

Figure 10: Noise robustness for the graph extraction procedure for various $\sigma$ field values.

**Field component weight** $\gamma$: The component weights may be thought of as an input normalization parameter. Therefore, we did not consider other values than $0.176$, which should normalize inputs to roughly values between $[0, 1]$.

**Resolution** $r$: In general, it is desired to have as low resolution as possible, as this will influence the input dimensionality and, correspondingly, the depth of the parametrizing NN for limited computational resources. At the same time, the accurate extraction of conformers and graphs must be ensured. Figure 11 shows how lowering the resolution from $0.33$ (our choice) to $0.44\text{Å}$ worsens "extraction accuracy" and "distance from original" across noise levels. We note, however, that $r = 0.44$ may still be a valid choice, assuming the resulting NN would have low enough amounts of noise.

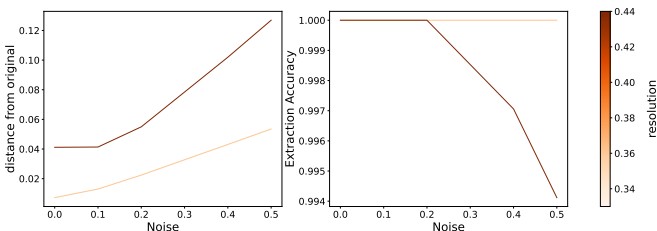

Figure 11: Noise robustness for the graph extraction procedure for two resolution levels.

## D  MODEL AND OPTIMIZATION PARAMETERS

**Diffusion Model**  We use 100 timesteps $t \in [0, 1]$ for our diffusion processes, the $\boldsymbol{\epsilon}$ parametrization, and the modified cosine scheduler for $\alpha_t$, as presented in main Section 4. The conditional scale for the classifier-free guidance was kept to $\beta = 2$ during the generation phase, for our main results (unless explicitly mentioned otherwise).

**Optimization** We optimize the $\mathcal{L}_{\text{simple}}$ objective using Adam optimizer with $8 \cdot 10^{-5}$ learning rate and $(0.9, 0.99)$ for the Adam's $\beta$ parameters. The models are trained using batch sizes of 32 and 64, for GEOM-Drugs and QM9 datasets, respectively.

**U-Net** We use a three-dimensional adaptation of the U-Net employed by (Ho et al., 2020). For both GEOM-Drugs and QM9 datasets, we use the same hyperparameters, except for the batch size. The architecture consists of three U-Net layers, with linear attention for each resolution level, except the last one, which uses full attention. Its initial convolution converts the 8 and 11 channels (for QM9 and GEOM-Drugs datasets, respectively), into a hidden representation of $\mathbb{R}^{128 \times H \times W \times D}$, which is halved for each of the $H, W, D$ dimensions, while its channel size is multiplied by 1, 2, and 3.

**Resources** For the QM9 experiments, we use four A100 GPUs (40GB memory version) for 180 hours, and 1.2 million iterations (or about 780 epochs). On the GEOM-Drugs dataset, we used the same four A100 GPUs for 330 hours and trained our explicit H GEOM-Drugs model for 1.32 million iterations (6 epochs), and our implicit one for 1.2 million iterations (5.5 epochs).

# E MOLECULE GRAPH EXTRACTION PARAMETERS AND ALGORITHMS

The molecular graph extraction has four components, used in the following order: deterministic peak extraction, atom positions optimization, deterministic bond extraction, and $\gamma_{ij}$ parameter optimization that removes false bonds.

**Deterministic peak extraction** The algorithm works by defining a simple, recursive flood-fill algorithm for retrieving the neighbors. Assuming the RBF functions have a small $\sigma$ parameter and reasonably small amounts of noise produced by the diffusion model, it should never fail to extract the correct number of atoms and their approximate positions from the fields.

---

**Algorithm 1** Peak extraction

---

    **Input:** fields $\mathbf{u} \in \mathbb{R}^{N_x \times N_y \times N_z}$, threshold $t$, $q \leftarrow \{\mathbf{x}; \mathbf{u}(\mathbf{x}) > t\}$; $A_{\mathbf{m}} \leftarrow \{\}$
    **repeat**
        $p = q.pop()$
        $neigh \leftarrow \text{get\_neigh}(p, t, \mathbf{u})$
        $A_{\mathbf{m}}.\text{insert}(\text{mean}(neigh, p))$
        $q.pop(neigh)$
    **until** q is empty
    **return** $A_{\mathbf{m}}$

---

**Optimizing the atom positions** The initial atom positions retrieved in the $A_{\mathbf{m}}$ sets are further optimized using Equation 13. Note how both atom numbers and their types are obtained using the peak extraction algorithm, and the optimization only fine-tunes their positions $\{\mathbf{m}_i\}$.

**Bond detection** The distances between any two bonds $b_{ij}$ and $b_{jk}$ are half as large as the distances between the atoms $a_i$ and $a_k$, meaning $||\mathbf{m}_{ij} - \mathbf{m}_{jk}||_2 = ||\mathbf{m}_i - \mathbf{m}_k||_2/2$. This may create a significantly larger overlap between the bond RBF components than the atoms' RBF components. Therefore, we treat the bond assignment differently and do not reuse Algorithm 1. Using the optimized atom positions $\mathbf{M}_a$, we check for all the atom pairs $\mathbf{m}_i, \mathbf{m}_j \in \mathbf{M}_a$ that are within some predetermined threshold distance $||\mathbf{m}_i - \mathbf{m}_j||_2 < d_1$ and create sets

$$B_{ij,b} = \{\mathbf{u}_b(\mathbf{x}); ||\mathbf{x} - \mathbf{m}_{ij}||_2 < d_2, \mathbf{u}_b(\mathbf{x}) > t_b, \mathbf{x} \in \mathcal{G}\},$$

where $\mathcal{G}$ consists of the evaluated field locations, $t_b$ is an additional bond threshold, and $d_2$ is the radius of the sphere centered on $\mathbf{m}_{ij}$, around which we check for $\mathbf{u}_b$ values. A covalent bond type $b$ is added between atoms $i$ and $j$ if $B_{ij,b} \neq \varnothing$.

To ensure we don't miss bonds due to the low resolution, we select a large sphere (determined by its radius $d_2$) around the approximate bond positions $\mathbf{m}_{ij}$ and small threshold values $t_b$. This may then recover bonds at positions where the model did not generate a true RBF bond component.

**Bond optimization** $\gamma_{ij}$    To remove erroneous bonds, extracted due to our lenient bond-extraction parameters $(d_1, d_2, t_b)$, we optimize the $\gamma_{ij}$ values for all $B_{ij,b} \neq \varnothing$ assuming the mean bond positions $\mathbf{m}_{ij}$ fixed, using equation 14. In Figure 12, we analyze how robust the method is to various degrees of uniform noise. We choose 300 molecules from the QM9 dataset and gradually increase the noise amount to the voxelized field values $\mathbf{u}$.

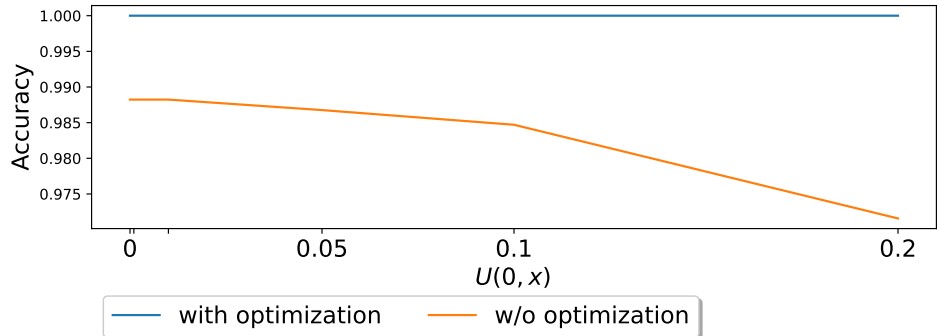

Figure 12: Performance of the graph extraction method for various degrees of noise. The bond optimization step is crucial and makes the method highly robust to noise.

Figure 13 illustrates a bond that was initially extracted based on $B_{ij,b} \neq \varnothing$, but removed after optimizing and finding the low $\gamma_{ij}$ value.

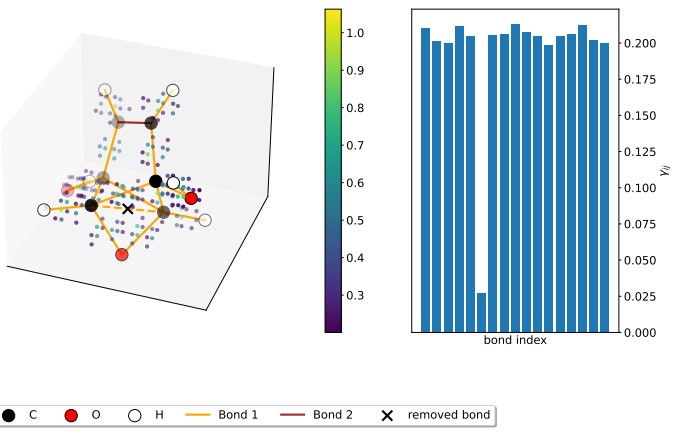

Figure 13: Upper left subfigure illustrates a noisy covalent bond type 1 channel values $\tilde{\mathbf{u}}_{b_1} = \mathbf{u}_{b_1} + \boldsymbol{\epsilon}$ (according to the color bar), where $\boldsymbol{\epsilon}$ has the same dimension as $\mathbf{u}_{b_1}$ and $\epsilon_{ijk} \sim \mathcal{U}(0, 0.1)$. The bond marked with "x" on the left is removed since its optimized $\gamma_{ij}$ value (right bar plot) is significantly lower than the rest.

**Parameters**    We choose threshold parameter $t = 0.3$ in Algorithm 1. For bond detection, we choose a large distance $d_1$ of $0.35$Å (for reference, the margin used to detect bond type 1 based on distance in the ablation Table 3 is $0.1$Å). The distance $d_2 = 0.45$ Å will result in the method checking for the closest midpoint voxel to $\mathbf{m}_{ij}$ and a few others directly in contact with it. The bond threshold is set to $t_b = 0.3$. Both the mean positions $\mathbf{m}$ and bond RBF component factors $\gamma_{ij}$ are optimized using gradient descent for 500 iterations and a learning rate of 5. All parameters were selected according to robustness analyses, such as the one illustrated in Figure 12, based on the training data.

# F GENERATED SAMPLES

## F.1 QM9 SAMPLES

Figures 14 and 15 depict a set of random valid samples from a model trained on QM9.

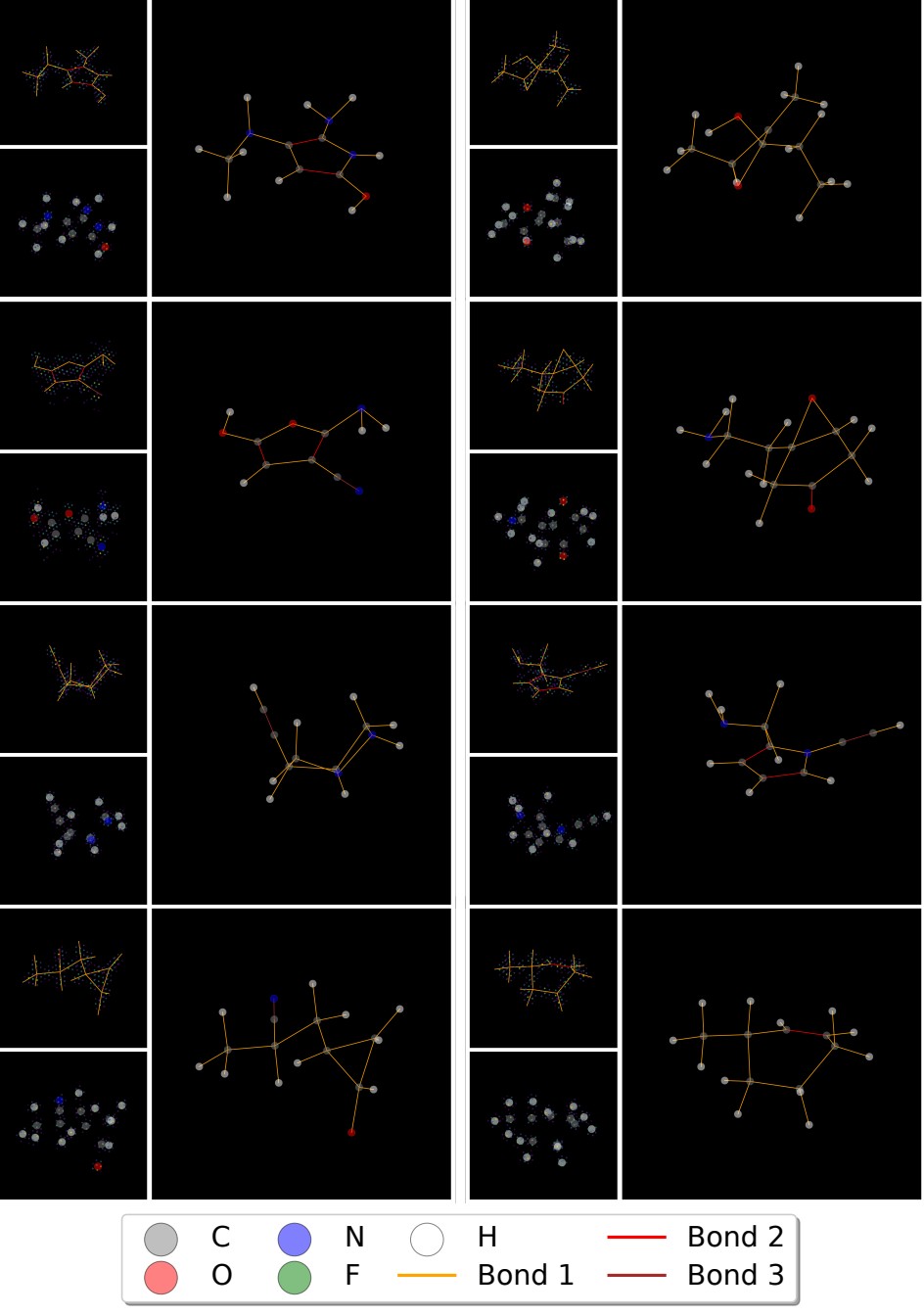

Figure 14: Random valid samples from FMG trained on QM9. To the left of each molecule, the top 200 largest $\mathbf{u}_a$ and $\mathbf{u}_b$ values are shown along their corresponding extracted atoms and bonds.

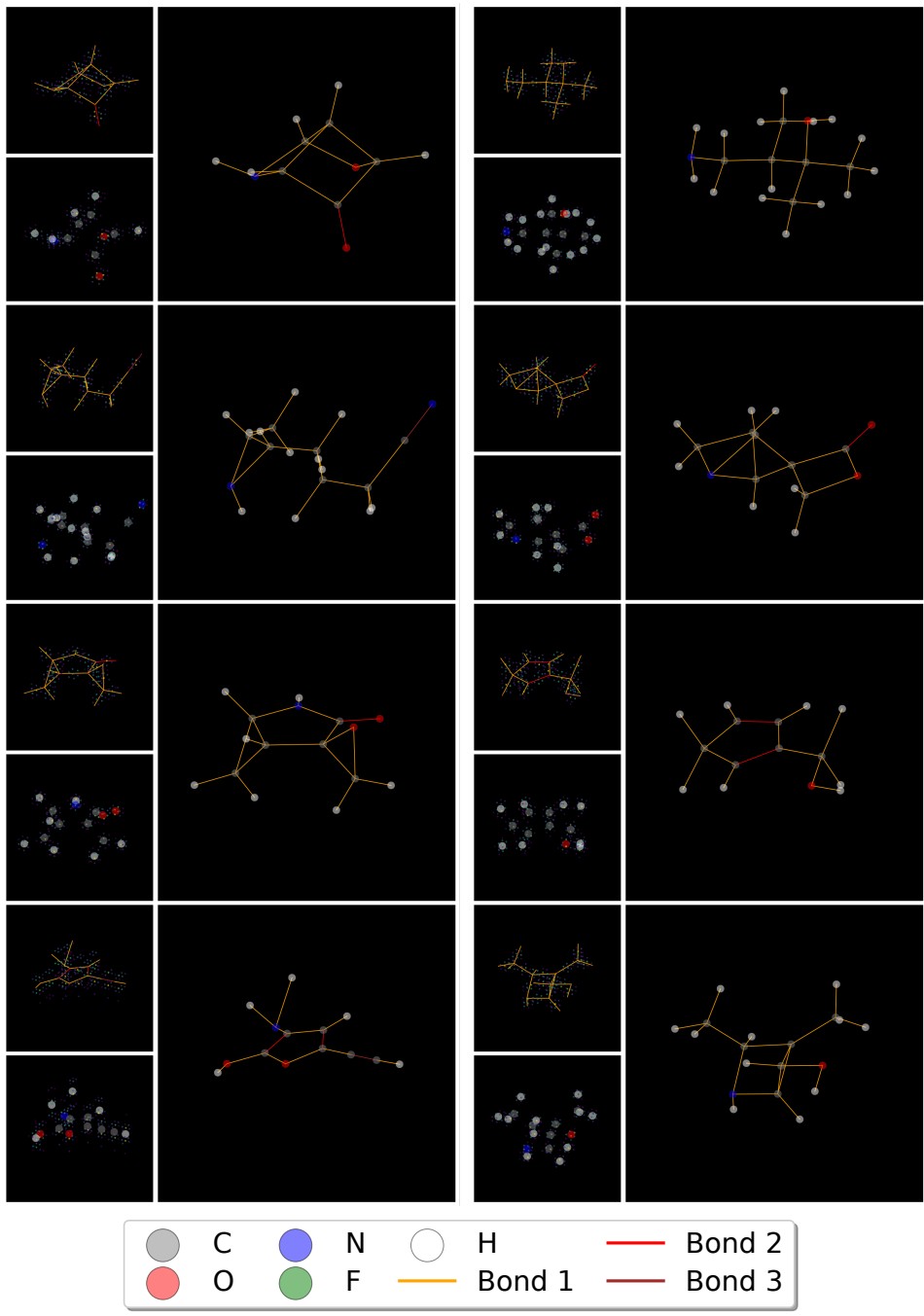

Figure 15: Random valid samples from FMG trained on QM9. To the left of each molecule, the top 200 largest $\mathbf{u}_a$ and $\mathbf{u}_b$ values are shown along their corresponding extracted atoms and bonds.

## F.2 GEOM-DRUGS SAMPLES

Figures 16 and 17 depict a set of random valid samples from a model trained on GEOM.

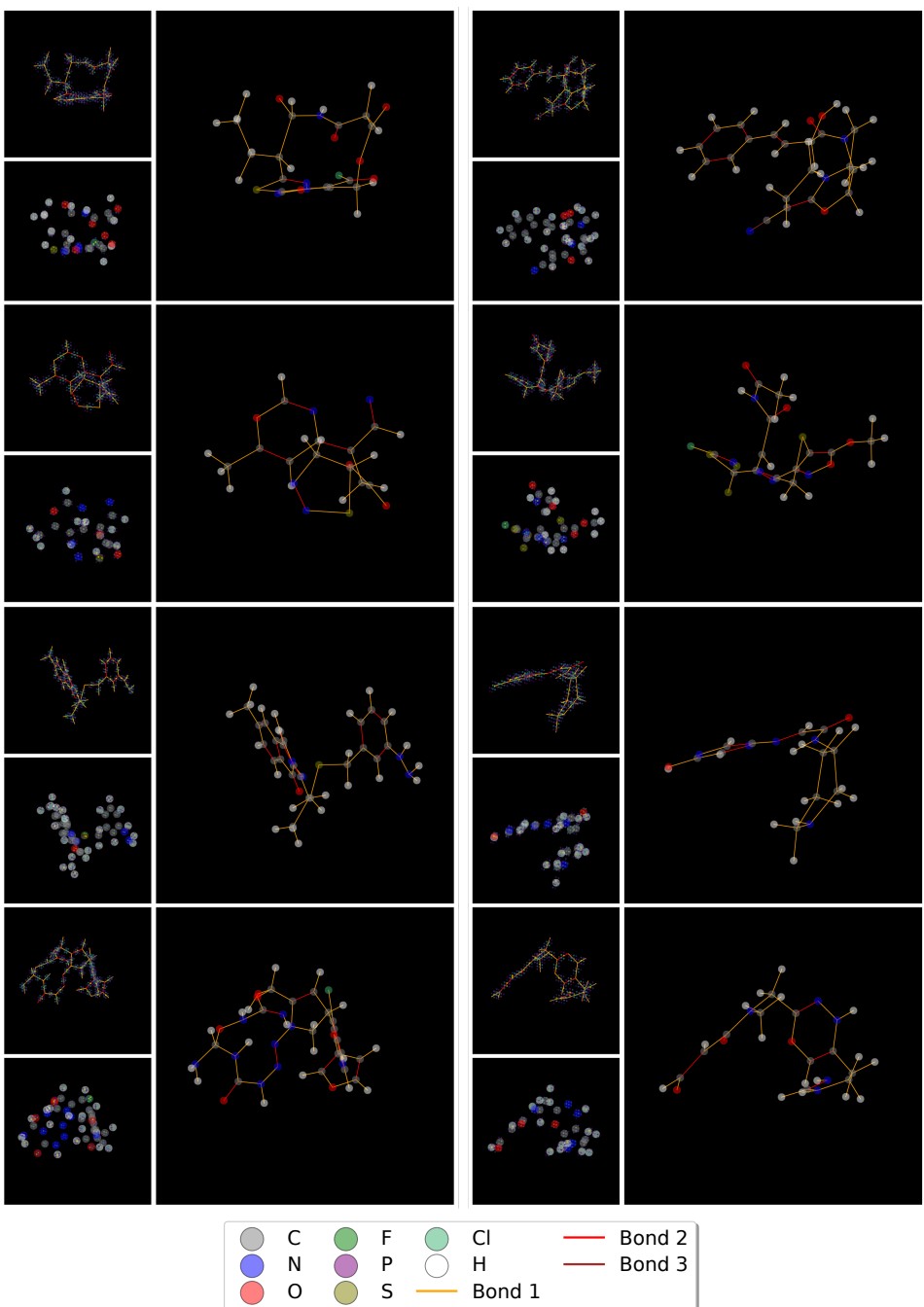

Figure 16: Random valid samples from FMG trained on GEOM. To the left of each molecule, the top 200 largest $\mathbf{u}_a$ and $\mathbf{u}_b$ values are shown along their corresponding extracted atoms and bonds.

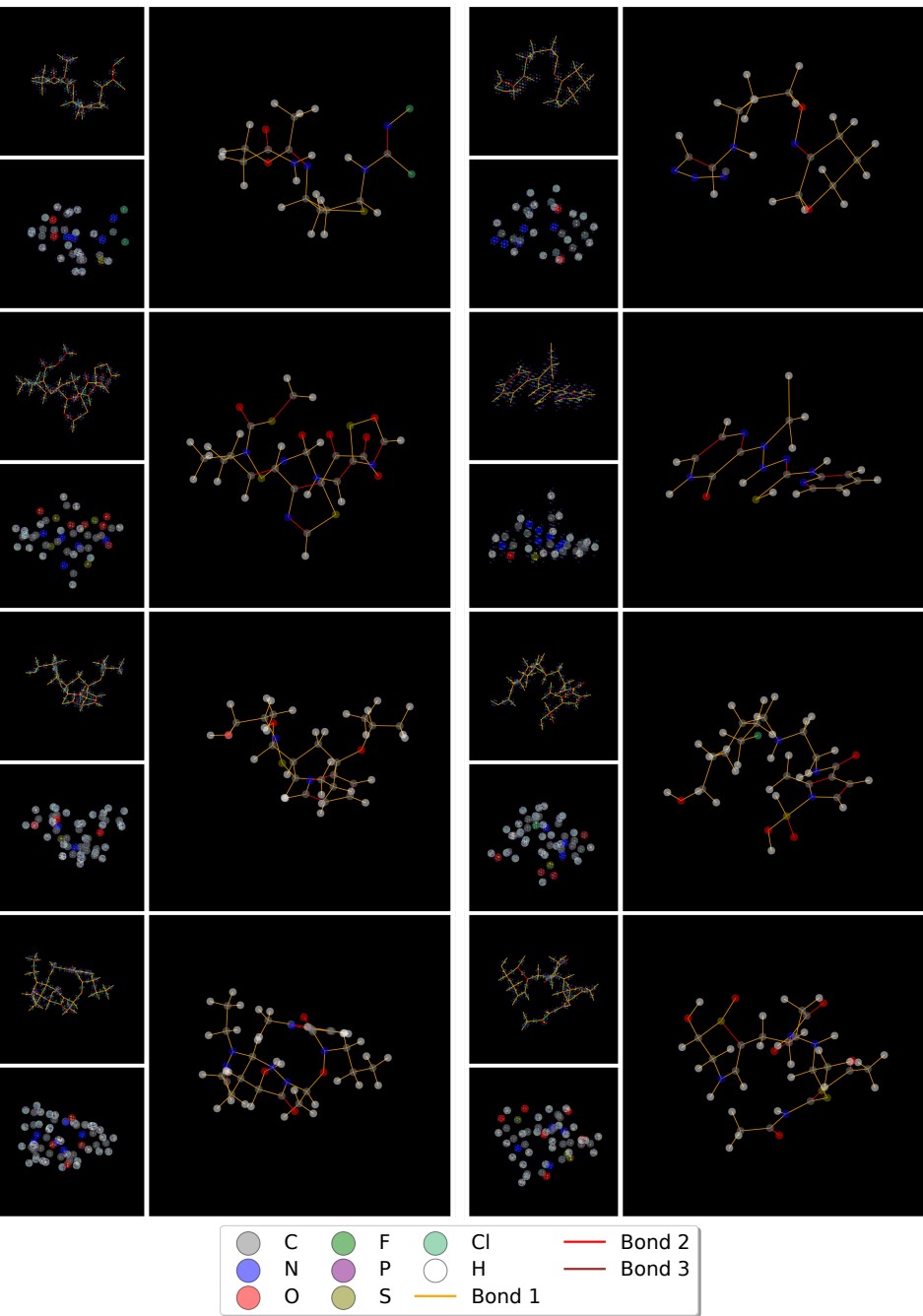

Figure 17: Random valid samples from FMG trained on GEOM. To the left of each molecule, the top 200 largest $\mathbf{u}_a$ and $\mathbf{u}_b$ values are shown along their corresponding extracted atoms and bonds.

# G  CLASSIFICATION-FREE GUIDANCE ANALYSIS

We create 17 and 91 bins for QM9 and GEOM-Drugs data sets, respectively, and assign them indices that we use as categorical conditioning variables. This led to a better match between the generated and data atom distributions (measured using $TV_a$) (see Table 3) and we deem it relevant when com-

paring to point-cloud-based approaches since these methods strictly enforce the number of generated atoms.

We determine how accurately the classification free guidance results in molecules containing the conditioned number of atoms. We compute the accuracy as the percentage of generated molecules in the correct conditioning bin $N$, the L1 distance between the binned number of atoms generated by the model $\tilde{N}$, and the conditioning bin $N$ as $\min(|\tilde{N}_{high} - N_{low}|, |N_{high} - \tilde{N}_{low}|)$ (or 0, when $N = \tilde{N}$), and "L1 lower" is the fraction of the total L1 distance attributed to incorrectly generated bins because of a strictly lower number of atoms generated $\tilde{N} < N$ (as a measure of distribution skewness).

In Figure 18, the generated bin accuracy is relatively low but is generally off by approximately 1 atom (measured by the L1 distance) for the QM9 dataset. In terms of the distribution shift, L1 lower suggests that the QM9 FMG model tends to generate more atoms than the conditioning variable.

Similar results in terms of accuracy and L1 distance can be noted in the GEOM-Drugs dataset (relative to the much larger possible number of atoms), but now the shift is in the opposite direction. This could be an artifact of the number of training iterations: early on in the QM9 dataset training experiments, we observed that the model tends to generate a lower number of atoms and bonds, which likely explains this opposite shift of the GEOM-Drugs method. For reference, our GEOM-Drugs model has been trained for about 6 epochs, compared to its QM9 counterpart (about 780).

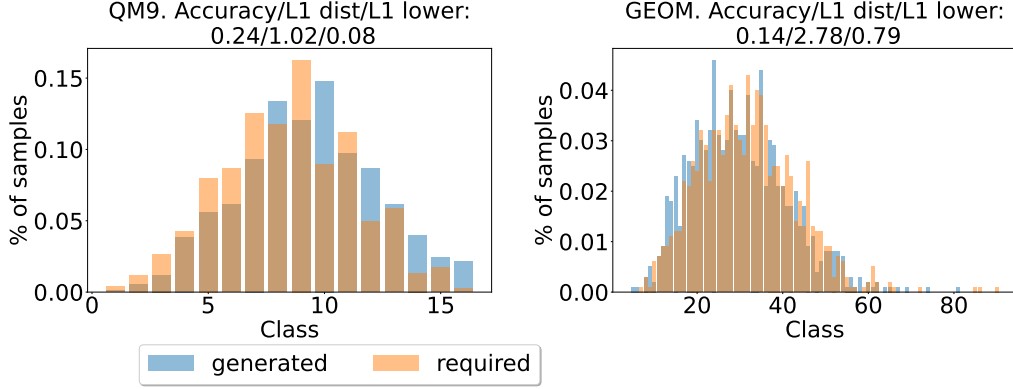

Figure 18: Histogram of the generated number of atom classes $\tilde{N}$ and the conditioning used $N$. Small and more significant atom number distribution shifts occur for QM9 and GEOM-Drugs datasets, respectively. Together with the result of Table 3 which shows how critical it is to enforce the correct number of atoms for $TV_a$, this result suggests that our method may remove atoms for stability, inducing an atom distribution shift.

## H    PERFORMANCE METRIC DETAILS

**Validity:**    Percentage of valid molecules, computed using RDKit's SanitizeMol function, computed using explicit H atoms.

**Neutrality:**    Percentage of generated atoms that have a formal charge of 0, and percentage of molecules that contain all atoms with a formal charge of 0.

Among our molecular quality metrics, the atom and bond total variation distances $TV_a$ and $TV_b$ are a good measure of how well most of the data modes are captured, whereas MST should better measure the quality of the molecules. For instance, if the method captures a single mode of distribution well, and generates very high-quality molecules, it may still have relatively bad $TV_a$ and $TV_b$.

**$TV_a$, $TV_b$**    We define $TV_a$ and $TV_b$ as:

$$\text{TV}_\text{a} = \sum_{a \in \mathcal{A}} \sum_{N_a = i}^{N} |p_\mathcal{D}(N_a) - p_g(N_a)|$$

$$\text{TV}_\text{b} = \sum_{b \in \mathcal{B}} \sum_{N_b = i}^{N} |p_\mathcal{D}(N_b) - p_g(N_b)|,$$

where $N_a$, $N_b$ are the numbers of atoms and bonds of type $b$, $a$ and $b$ are the atom and bond types, $\mathcal{A}$ and $\mathcal{B}$ are the sets of atoms and bonds considered, $p_\mathcal{D}$ and $p_g$ are probability distributions based on data and generated statistics, and $|\cdot|$ is the absolute value.

**MST:** Denoting the Tanimoto Similarity as TS$= (\mathbf{v}_1 \cap \mathbf{v}_2)/(\mathbf{v}_1 \cup \mathbf{v}_2)$, where $\mathbf{v}_1$ and $\mathbf{v}_2$ are Morgan Fingerprints based on molecular graphs, we compute the average maximum similarity to test (MST) as:

$$\frac{1}{N} \sum_{i=1}^{N} \max_{\mathbf{v}_t \in \mathcal{V}_\text{test}} \text{TS}(\mathbf{v}_i, \mathbf{v}_t),$$

where $\mathcal{V}$ is the set of all test set Morgan Fingerprints.

**BL, BA, CE$_\text{MMFF}$, and CE$_\text{xTB}$:** We determine the Wasserstein $W_1$ distance between the generated and data bond angles, bond lengths, and Merck molecular force field energies (in kcal/mol).

The angle distributions are defined for all atom types that have two or more bonds, and we consider all pair-wise angles between position vectors $\mathbf{m}_i - \mathbf{m}_k$, and $\mathbf{m}_j - \mathbf{m}_k$, for any two atoms $i$ and $j$ connected to the same central atom $k$. We consider the average $W_1$ distance over all atom types, and we denote it by BA.

Similarly, we create bond Euclidean distance distributions for all bond types $b \in \mathcal{B}$ and denote our averaged $W_1$ distance over all bond types as BL.

We use discretized bins of length $l_\theta = 0.1$ degrees for angles, $l_d = 0.01$Å for bond Euclidean distances, and $l_e = 0.1$ (kcal/mol) for conformation energies, and approximate the generated and data cumulative distribution functions $F_g$ and $F_d$ and compute:

$$\text{BA} = l_\theta \frac{1}{N_\mathcal{A}} \sum_{a \in \mathcal{A}} \sum_{x \in B_a} |F_{d,a}(x) - F_{g,a}(x)|$$

$$\text{BL} = l_d \frac{1}{N_\mathcal{B}} \sum_{b \in \mathcal{B}} \sum_{x \in B_b} |F_{d,b}(x) - F_{g,b}(x)|,$$

where the bin upper limits $B_a$, $B_b$, and $B_e$ are chosen to contain all maximum and minimum values of the generated and training data samples, $N_\mathcal{A}$ and $N_\mathcal{B}$ are the number of unique atoms and bonds considered (e.g., we excluded H and F for BA computations, as these atoms never have two bonds).

For both Merck molecular force field (CE$_\text{MMFF}$) and xTB (CE$_\text{xTB}$) we use the same bin length $l_e$ and compute the metics in the same way:

$$\text{CE} = l_e \sum_{x \in B_e} |F_{d,e}(x) - F_{g,e}(x)|,$$

where $F_{d,e}$ may be either MMFF or xTB CDF of generated and data energies.

**Outlier problems of $W_1$ distance:** Compared to our method, point-cloud methods may generate a molecule with an arbitrarily long bond length (i.e., $l \to \infty$), making the BL $W_1$ metric arbitrarily high based on one or few bonds. Let $p_\theta$ and $p_\mathcal{D}$ be the generated and data bond length distributions. If their corresponding CDFs are $F_\theta$ and $F_\mathcal{D}$, the $W_1$ distance can be equivalently written as

$$W_1(p_\theta, p_\mathcal{D}) = \int_\mathbb{R} |F_\theta(l) - F_\mathcal{D}(l)| dl$$

where the integration is over the generated and data molecule lengths $l$. In fields, bond lengths are constrained by the chosen evaluated points $\mathcal{G}$, while in point clouds, $l$ may be arbitrarily high. In turn, this can theoretically incur arbitrarily high penalties due to a few outliers. In practice, this does not happen (Appendix A.1, Figure4, depicting the CDFs of bond lengths), and we further ensure we don't have such an outlier advantage by restricting the bond lengths and angles to only those generated within the data limits when computing BA and EL metrics.

## I   EXPLICIT AROMATIC BONDS AND NEUTRALITY

**Problem explanation**   In our settings, we have used the Kekulé molecule representations, which represent aromatic bonds as alternating covalent bond types 1 and 2. In Figure 19, we show three different such aromatic molecules, using the Kekulé representation. Noting that the N and C valencies are 3 and 4, respectively, all formal charges are 0 in these cases (all carbons are assumed to be implicitly connected to a H).

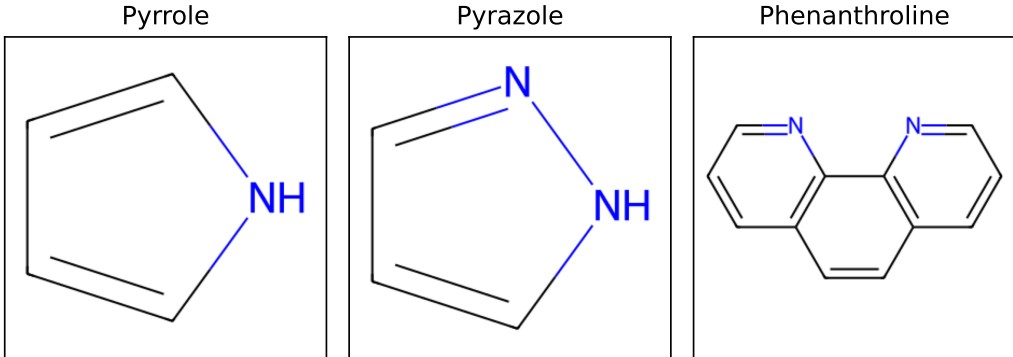

Figure 19: Three aromatic molecules, with increasingly harder to determine neutrality.

Instead of the alternating bond types 1 and 2, an equivalent representation denotes these bonds as a separate "aromatic" bond. By considering an electron occupancy of 1.5 for aromatic bonds, Pyrrole has the problem that the N in its molecular graph is apparently (but not in actuality) ionized, having a formal charge of -1. Furthermore, in Pyrrazole, only one of the N atoms should have the (apparent) formal charge -1. Lastly, the C atoms connecting to two rings in Phenanthroline would have a formal charge of -0.5 (as it connects with three 1.5 bonds).

This induces ambiguity. One way to solve this would be to use RDKit and convert the explicit aromatic representation to Kekulé representations. However, in the case of invalid molecules (about 38% of the generated molecules by MiDi), it becomes hard, if not impossible, to determine exactly how many atoms connecting to a (potentially incorrect) aromatic ring have the formal charge 0.

**Explaining the reported results for MiDi**   We have used the molecules reported in the MiDi github repository. We note that on the QM9 dataset, both our method and MiDi generate molecules in Kekulé form, and they are directly comparable in Table 1 (QM9 columns). For the GEOM-Drugs neutrality results in Table 1 (GEOM-Drugs columns), we have reported an upper-bound for MiDi, by considering bond type "aromatic" as 1.5 electron occupancy and allowing C to have both 0 and -0.5 formal charge. Since many N atoms in GEOM-Drugs are (in actuality) ionized, it is not "allowed" to have a formal charge of -1. Regardless, the comparison is very hard to do, and it is difficult to say which model generates more neutral atoms or molecules.

**MTS bias** We compute the maximum test similarity using valid molecules. This results in a bias in the methods that generate aromatic bonds explicitly, as RDKit will require these molecular graphs to be converted to Kekulé representations (Table 2). In this case, this metric will not consider molecules containing incorrect aromatic rings (as determined by RDKit's SanitizeMol). However, in our case, since we directly generate Kekulé molecule representations, this quality filter does not exist anymore, and samples that have incorrect aromatic rings (but are not identified as aromatic because of e.g., one circle atom being incorrectly hybridized) will be considered in the final metric. Note how this bias is unrelated to the performance of the generative model, as those unkekulizable molecules are still generated by explicit-aromatic generating methods, but are simply filtered out by this metric.

## J    COMPLEXITY ANALYSIS

### J.0.1    TIME SCALING

We use convolutional layers as the main blocks of our UNet parametrization. As such, the complexity of our NN scales linearly with $H \cdot W \cdot D$. We omit the channel term $C_{\text{in}} \cdot C_{\text{out}}$, as it is equivalent to the hidden activations on node states from point-cloud methods. Linear attention layers are added after each UNet layer, giving $\mathcal{O}(H \cdot W \cdot D)$.

Comparatively, all layers of point-cloud DDPM methods scale quadratically in the number of atoms $\mathcal{O}(N^2)$. The scaling of these two methods is not directly comparable, since the number of atoms does not directly relate to the necessary volume that encompasses all molecules of that size, which is highly dataset-dependent. Still, a theoretical upper limit can be determined.

Consider a dataset that contains molecules of atoms, which, for simplicity, are all C atoms bonded through covalent bond type 1 (approximately Å). We now discuss how the atoms should be arranged to create the worst possible scenario (without considering physical constraints). The upper limit of our grid sizes would be achieved when all three of the following n-atom molecules are present:

- One molecule has all atoms placed linearly, which will determine $H = N \cdot 1.35$Å, and $W = D = 0$.
- One molecule is perfectly planar, having $H = W = \frac{N \cdot 1.35}{2}$ and D=0. Note that $W > H$ is not possible, since our alignment method (Section 3.5) would rotate the molecule s.t. the longest axis is $H$.
- One molecule has $H = W = D = \frac{N \cdot 1.35}{3}$

The scaling of our convolutional layers would be $\mathcal{O}(\frac{1.35N}{r} \cdot \frac{1.35N}{2r} \cdot \frac{1.35N}{3r})$, where $r$ is the resolution of our evaluated field locations. We compare the number of floating point operations in terms of the number of atoms in Figure 20. For FMG, we show both the theoretical maximum and the grids that encompass GEOM-Drugs molecules of various numbers of atoms. In Figure 21, we determine the training and generation time (in seconds) of FMG and EDM.

When close to the theoretical upper limit, our method becomes intractable, but this case is highly unrealistic (Figure 20, comparing "FMG max" with "FMG GEOM"). In Figure 2 a), FMG has a comparable generation time but takes longer to train. Figure 2 b) depicts our training time for various molecule sizes, where 99.7% of the molecules are kept (used in this work). The molecule graphs were chosen s.t. they fit within a grid size of $64 \cdot 40 \cdot 32$.

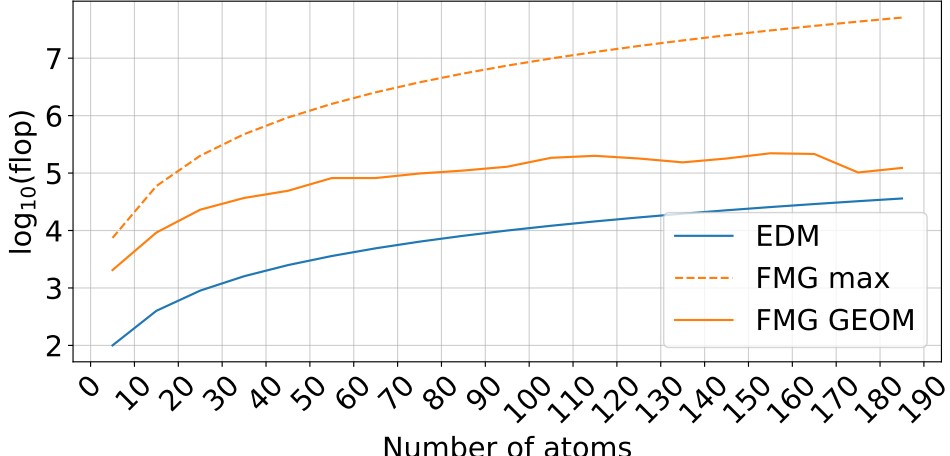

Figure 20: Number of floating point operations for EDM and FMG over the number of atoms in a molecule. The theoretical maximum of FMG differs substantially from the real datasets, and its practicality relies on application specifications. For the axis lengths in FMG 0.33Å was used (same value as the one used in our model).

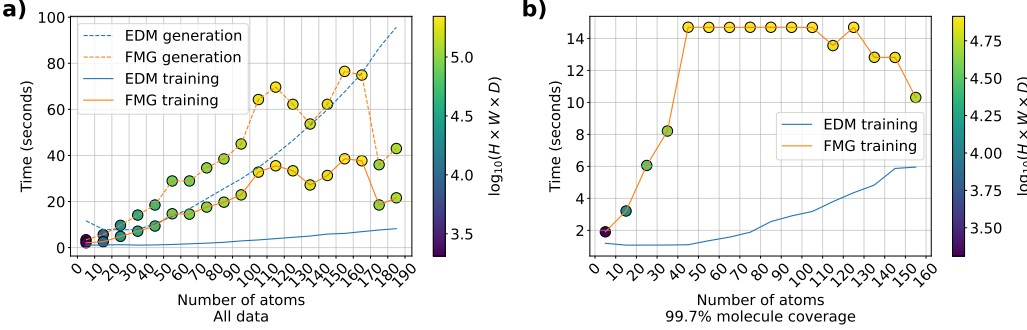

Figure 21: Time elapsed (in seconds) for EDM and FMG models, over various molecule sizes (in number of atoms, and tensor field volumes, for EDM and FMG, respectively). We used 100 iterations with a batch of four molecules to determine the training curves. For this experiment, we used an Nvidia A100 GPU. a) All GEOM-Drugs molecules are considered. b) Only molecules used in the current work (99.7% of the molecules) that fit within our grid size of 64·40·32 are used in the analysis.

### J.0.2  MEMORY SCALING

Increasing the molecule sizes has a significant impact on memory overhead. Similar to the analysis done in the time scaling analysis in Appendix J.0.1, we construct grids that can fit all GEOM-Drugs molecules with a certain number of atoms $N$. We report the maximum memory allocated for EDM (Hoogeboom et al., 2022) and our method, when generating samples of various atom sizes (determined on the x label by the number of atoms $N$, or dots denoting volume sizes), with 32 samples per batch.

Figure 22 a) shows that our method performs better for larger molecules, but worse for smaller ones. We note here that GNN architecture batching is done by considering a single large graph, where nodes from different batches are disconnected. Therefore, adjacency matrices of this implementation

have poor scaling for large batch sizes: $((\sum_{b \in B} N_b)^2)$, where $B$ is the batch sizes and $N_b$ is the number of atoms in a molecule.

We find that increasing the number of unique channels (atoms or bonds) is negligible and linear. The additional memory required stems from the input CNN layer's kernel dimension (the depth sizes of the initial 3D CNN kernels are equal to $|\mathcal{A}| + |\mathcal{B}|$), and the larger intermediate states. However, these are considerably smaller than the architecture's overhead (Figure 22 b). Channel scaling is performed with a batch size of 1.

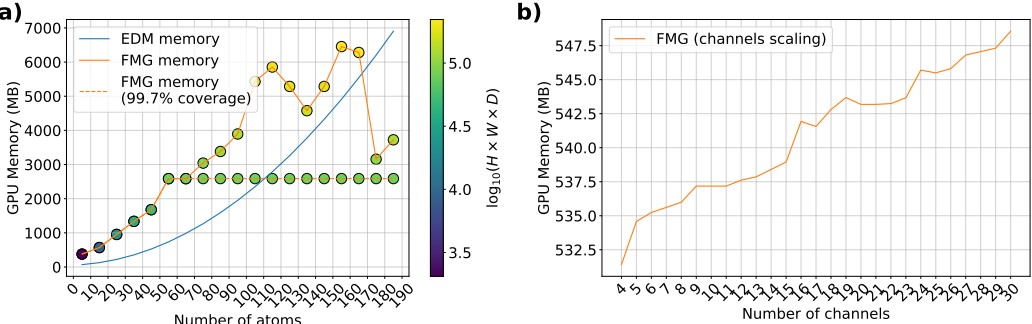

Figure 22: Memory scaling in terms of molecule size for generating molecules. **a)** We report the maximum GPU memory allocated when sampling for various grid sizes, which fit all molecules of the corresponding number of atoms. We additionally report the memory when using the grid sizes chosen for this work. **b)** Generation memory in terms of the number of channels.

