# OpenReview forum: "E(3)-equivariant models cannot learn chirality: Field-based molecular generation"
_ICLR.cc/2025/Conference — ICLR 2025 Poster_

### Official Review · Reviewer_hRsL · 2024-10-29

**Soundness:** 2
**Presentation:** 3
**Contribution:** 3
**Rating:** 5
**Confidence:** 4

**Summary:**

The paper proposes a field-based diffusion method for molecule generation that is not invariant with respect to the chirality of the generated molecules. Recognizing that chirality is a critical property influencing the bioactivity of molecules—with several famous examples underscoring its impact on molecular toxicity and drug efficacy—the approach addresses an important aspect of molecular design.

The results demonstrate comparable performance to EDM and MiDi in terms of molecule validity and stability. The authors also highlight molecular neutrality as an important metric for generated molecules. Furthermore, they assess MMFF energies for the 3D molecules and compute the Wasserstein distance between the bond length distributions and bond angle distributions of the generated molecules and those of the dataset molecules.

**Strengths:**

The authors proposed a novel approach to molecular generation using field representations and developed a required theoretical background for the approach. The authors prove challenges to incorporate the same restriction in current E(3) equivariant models.

**Weaknesses:**

1. Overstated Claims of State-of-the-Art Performance: Although the paper claims to achieve state-of-the-art (SOTA) performance, it falls significantly behind previously published models such as JODO, EQGAT-Diff, and the more recent Semla-Flow—all of which were published before ICLR 2025. The performance gap is quite substantial. For larger and more realistic molecules in the GEOM Drugs dataset, only 16% have all valid atoms according to the paper's model, whereas other models are approaching perfect scores on these metrics.

2. Equivariance Requirement Not Essential for High Performance: Some models that operate directly in E(3) space are eliminating the need for equivariance. For instance, Simplified Scalable Conformer Generation (MCF) demonstrates significant performance improvements in conformer generation without the requirement for equivariance. While MCF tackles a slightly different task, it illustrates that we can develop equivariance-unaware models for molecular modeling that achieve genuine SOTA performance.

3. Incorrect Basis for Neutrality Metrics: The neutrality metrics in the paper are based on inaccurate assertions. Specifically, the statement: "Atom neutrality is the percentage of atoms with a formal charge of 0, while molecules are neutral if all their atoms are neutral." is incorrect. For example, nitromethane has charged oxygen and nitrogen atoms but is an overall neutral molecule. Atoms within a molecule can carry formal charges that balance out, resulting in a neutral molecule despite individual atoms being charged.

4. Inadequate and Incomplete 3D Structure Evaluation: The evaluation of the 3D structures is poor and incomplete. The paper only compares distribution differences for bond lengths, angles, and energies. These benchmarks are difficult to interpret and can be significantly affected by a single outlier result. Depending on the implementation—for example, in MiDi's implementation—one problematic molecule can drastically alter the results. Even with these limitations, the model does not outperform MiDi on these benchmarks.

5. Overstated Importance of Chirality Awareness in the Model: The emphasis on chirality awareness in the model may be overstated. 3D molecule generation serves as a foundational model for structure-based drug design or molecular docking. In these applications, the provided environment (such as a protein or part of a protein) creates asymmetric restrictions on the generated molecule, naturally leading to the production of the appropriate enantiomer. Additionally, in conditional generation, the classifier itself can learn chirality in a simpler way.

**Questions:**

Questions to the Authors:

1. Your model underperforms compared to JODO, EQGAT-Diff, and Semla-Flow but claims SOTA performance. Can you add a comparison to one of these models?

2. Since non-equivariant models like MCF achieve high performance, can you check whether it still cannot distinguish chiral conformers?

3. Your neutrality metric seems inaccurate (or at least description) (e.g., nitromethane is neutral despite charged atoms). Can you address this issue?

4. Is the strong focus on chirality awareness justified when environmental factors often determine enantiomer generation in practice?

---

> ### Author Response · Authors · 2024-11-19
>
> We thank the reviewer for their time and effort in reviewing our work. Below, we address the raised questions.
>
> **Note: please refer to the last rebuttal comment for references**
>
> **Weakness 1** Overstated Claims of State-of-the-Art Performance: Although the paper claims to achieve state-of-the-art (SOTA) performance, it falls significantly behind previously published models such as JODO, EQGAT-Diff, and the more recent Semla-Flow—all of which were published before ICLR 2025. The performance gap is quite substantial. For larger and more realistic molecules in the GEOM Drugs dataset, only 16\% have all valid atoms according to the paper's model, whereas other models are approaching perfect scores on these metrics. **Question1** Your model underperforms compared to JODO, EQGAT-Diff, and Semla-Flow but claims SOTA performance. Can you add a comparison to one of these models?
>
> *Stability/neutrality difference:* We respectfully disagree - the "stability" metric reported by Jodo [1], EQGAT-DIFF [2], and SemlaFlow[3] does not correspond to the "neutrality" metric we report. Those works consider charge $\ne 0$ atoms as stable, whereas our "neutrality" definition only considers 0-charge to be "neutral". Please refer to "Weakness 3" section for details on the "neutrality"  metric.
>
> *Differences in GEOM benchmark:* We also note the GEOM benchmark discrepancy: [1] uses only the lowest energy conformation, [5] uses the lowest 5 energy conformations and we use the lowest 30, similar to [4]. Removing higher energy conformations should improve our performance, but it is difficult to speculate by how much, and therefore hard to compare on GEOM-drugs.
>
> *Neutrality metric for aromatic methods compared to Kekule (ours):* Finally, please also note how we compute neutrality in explicit aromatic settings (remark ($\ddagger$)), where C has both valencies 4 and 4.5 considered correct (Appendix I).
>
> Below we report JODO [1], our method, and MiDi [5] results (for context). Our method still achieves the best or highly competitive performance across metrics, with a disadvantage on TV\textsubscript{a}, which likely stems from the lack of hard-constrained atom count generation (ablation on $\beta$, Table 3 of main text). We now add JODO to our benchmark tables.
>
> *QM9:*
>
> | Model | R  | Val        | Neutr (mol) | Neutr (atm) | TVa (↓)       | TVb (↓)       | MST (↑)       | BL (↓)        | BA (↓)        |
> |-------|----|------------|-------------|-------------|---------------|---------------|---------------|---------------|---------------|
> | MiDi  |    | 96.7       | 84.5        | 98.3        | 0.19          | 0.81          | 0.57          | 1.92          | 0.95          |
> | JODO  |    | 98.1       | 88.2        | 98.5        | 0.19          | 0.89          | 0.58          | 0.62          | 0.83          |
> | FMG   |    | 98.8 ± 0.3 | 91.5 ± 1.4  | 99.4 ± 0.1  | 0.52 ± 0.02   | 0.95 ± 0.01   | 0.54 ± 0.00   | 0.59 ± 0.01   | 1.08 ± 0.12   |
> | Data  |    | 99.6       | 99.9        | 99.6        | -             | -             | -             | -             | -             |
>
> *GEOM:*
>
> | Model | R          | Val  | Neutr (mol) | Neutr (atm) | TVa (↓) | TVb (↓) | MST (↑) | BL (↓) | BA (↓) |
> |-------|------------|-------|-------------|-------------|----------|----------|----------|----------|----------|
> | MiDi  | (‡, *)     | 62.7  | 31.3        | 96.8        | 1.07     | -        | 0.53     | -        | 1.38     |
> | JODO  | (‡, *)     | 77.8  | 36.7        | 97.0        | 1.07     | -        | 0.58     | -        | 0.35     |
> | FMG   |            | 64.9  | 16.5        | 94.7        | 2.44     | 4.78     | 0.49     | 2.74     | 1.00     |
> | Data  |            | 89.6  | 86.2        | 99.4        | -        | -        | -        | -        | -        |
>
>
> **Weakness 2** Equivariance Requirement Not Essential for High Performance: Some models that operate directly in E(3) space are eliminating the need for equivariance. For instance, Simplified Scalable Conformer Generation (MCF) demonstrates significant performance improvements in conformer generation without the requirement for equivariance. While MCF tackles a slightly different task, it illustrates that we can develop equivariance-unaware models for molecular modeling that achieve genuine SOTA performance.
>
> We agree that eliminating architecture invariance constraints in favor of model scaling is a promising future direction and thank the Reviewer for the reference supporting our work. However, E(3) modeling is well-motivated as it encodes symmetry constraints of 3D objects and therefore guarantees to obtain similar representations for similar molecules, and has therefore been employed in many recent works [4,5,6,7,9]. We, therefore, deem our theoretical contributions highly valuable, in the context of molecules, tackling: i) the lack of chirality-awareness in E(3) modeling; ii) intractable chiral-aware extension of E(3) methods (Section 3, *Chirality and isometry groups*).

---

> > ### Author Response · Authors · 2024-11-19
> >
> > **Question 2** Since non-equivariant models like MCF achieve high performance, can you check whether it still cannot distinguish chiral conformers?
> >
> > Thank you for your suggestion. However, since methods like MCF model $p(C|G)$ (conformer $C$ given graph $G$), this falls outside the scope of the current work of novel compound generation $p(C,G)$. We also point out that chiral awareness while keeping rotational invariance in diffusion point-cloud methods is tractable in conformer generation ($p(C|G)$) [8], compared to novel compound generation $p(C,G)$, where we investigate the computational challenges of introducing chiral-awareness for E(3) methods.
> >
> > **Weakness 3** Incorrect Basis for Neutrality Metrics: The neutrality metrics in the paper are based on inaccurate assertions. Specifically, the statement: "Atom neutrality is the percentage of atoms with a formal charge of 0, while molecules are neutral if all their atoms are neutral." is incorrect. For example, nitromethane has charged oxygen and nitrogen atoms but is an overall neutral molecule. Atoms within a molecule can carry formal charges that balance out, resulting in a neutral molecule despite individual atoms being charged. **Question 3** Your neutrality metric seems inaccurate (or at least description) (e.g., nitromethane is neutral despite charged atoms). Can you address this issue?
> >
> >
> > We thank Reviewer for noting this. We agree that our metric's definition does not correspond to the physical one, and now mention this in Section 5. Our motivation for naming this metric "neutrality" instead of stability stems from the literature's ambiguity on the "stable" metric definition.
> >
> > *Motivation of the name "neutrality":* "Stability" was first introduced with the same definition as our "neutrality" definition in [4,7,9], but has since been adapted in various works with different definitions: [6] modifies N and C charges (post-generation) until it finds an RDKit-sanitizable molecule or fails within a predefined number of timesteps, [5] predicts charges and compares them to the true one, and [1] defines a table of "correct valencies", without explicitly modeling charges. While these approaches are perfectly reasonable in deployment settings, we argue that they make benchmarking difficult and therefore opted for the definition used in this work, and renamed it "neutrality".
> >
> > **Weakness 4.1** Inadequate and Incomplete 3D Structure Evaluation: The evaluation of the 3D structures is poor and incomplete. The paper only compares distribution differences for bond lengths, angles, and energies. These benchmarks are difficult to interpret and can be significantly affected by a single outlier result. Depending on the implementation—for example, in MiDi's implementation—one problematic molecule can drastically alter the results.
> >
> > We chose the BA and BL metrics since they have been utilized in prior work [5,10], to which we added Merck molecular force field energies W$\_1$ (EC). We now briefly mention the outlier concern in Section 5, and elaborate in Appendix H. We agree that outliers could, in principle, unevenly affect point-cloud methods, as point-clouds have unconstrained bond lengths, but this is generally not a big issue in practice (Figure 4, Appendix A.1, depicting the CDFs of bond lengths). We also recomputed conformation metrics by excluding outliers (restricting generated bond lengths and angles to dataset limits), and the results did not change significantly, and preserved the original model order:
> >
> > | Model | QM9 BL (↓) | QM9 BA (↓) Val | GEOM BL (↓) | GEOM BA (↓) |
> > |-------|------------|----------------|-------------|-------------|
> > | EDM   | 0.47       | 0.68           | 8.84        | 4.36        |
> > | MiDi  | 1.92       | 0.95           | -           | 1.38        |
> > | FMG   | 0.59 ± 0.01 | 1.08 ± 0.12    | 2.74        | 1.00        |
> >
> > Appendix H, below the BL/BA/EC metric description now contains: Compared to our method, point-cloud methods may generate a molecule with an arbitrarily long bond length (i.e., $l\rightarrow \infty$), making the BL W$\_1$ metric arbitrarily high based on one or few bonds. Let $p\_{\theta}$ and $p\_{\mathcal{D}}$ be the generated and data bond length distributions. If their corresponding CDFs are $F\_{\theta}$ and $F\_{\mathcal{D}}$, the $W\_1$ distance can be equivalently written as
> >
> > $$ W\_1(p\_{\theta},p\_{\mathcal{D}}) = \int\_{\mathbb{R}}|F\_{\theta}(l)-F\_{\mathcal{D}}(l)|dl$$
> >
> > where the integration is over the generated and data molecule lengths $l$. In fields, bond lengths are constrained by the chosen evaluated points $\mathcal{G}$, while in point clouds, $l$ may be arbitrarily high. In turn, this can theoretically incur arbitrarily high penalties due to a few outliers. In practice, this does not happen (Appendix A.1, Figure 4), and we ensure we don't have such an outlier advantage by restricting the bond lengths and angles to only those generated within the data limits when computing BA and EL metrics.

---

> > > ### Author Response · Authors · 2024-11-19
> > >
> > > **Weakness 4.2** Even with these limitations, the model does not outperform MiDi on these benchmarks.
> > >
> > > We note that we took further precautions to ensure that point-cloud methods' metrics aren't affected by outliers. Further, we argue that, especially on QM9, our BL W$\_1$ result of 0.0059Å and BA W$\_1$ of 1.08 (°), are well within acceptable margins - Table 3 shows that bond types extracted from molecular conformations alone (FMG$\_{dist}$) achieves highly competitive performance.
> > >
> > >
> > > **Weakness 5.** Overstated Importance of Chirality Awareness in the Model: The emphasis on chirality awareness in the model may be overstated. 3D molecule generation serves as a foundational model for structure-based drug design or molecular docking. In these applications, the provided environment (such as a protein or part of a protein) creates asymmetric restrictions on the generated molecule, naturally leading to the production of the appropriate enantiomer. Additionally, in conditional generation, the classifier itself can learn chirality in a simpler way. **Question 4.** Is the strong focus on chirality awareness justified when environmental factors often determine enantiomer generation in practice?
> > >
> > >
> > > **Chirality importance:** We agree that environmental factors determine optimal 3D configurations. Enantiomer configuration affects a ligand's binding affinity to a protein, as Reviewer points out. However, chirality influences more factors: metabolization, (enzymes metabolizing drugs are often chiral), chemical reactivity, and physical properties. Capturing all molecule distribution properties will therefore not be possible with chiral-unaware models. We argue that this is particularly crucial when modeling drug-like datasets: for example, a chiral-unaware method is very unlikely to generalize the concept of "metabolization". Therefore, chirality deserves attention from the community.
> > >
> > >
> > > *Chirality-awarness through classifiers:* We also have to disagree that chirality awareness in conditional generation settings is easily attainable unless the conditioning variables contain the molecular graph  $p(C|G, y)$ (conformation $C$, graph $G$, optional property $y$), which is outside the scope of novel compound generation ($p(C,G)$). We consider two ways of utilizing $p(y|C,G)$:
> > >
> > > * *Filtering samples from chiral-unaware $p\_\theta(C,G)$ with a chiral-aware classifier $p(y|C,G)$*: We argue that chiral aware $p\_\theta(C,G)$ would better model "drug-like" (e.g., metabolizable) drugs, than a chiral-unaware model. If $p(y|C,G)$ is a ligand affinity classifier to a protein $P$ (i.e., $p(y|C,G,P)$), then we would only filter out non-binders, and not be able to discriminate drug-like from non-drug-like molecules.
> > >
> > > * *Diffusion Classifier guidance with a chiral-aware classifier $p(y|C,G)$*: Chiral awareness (with rotational invariance) is simpler in discriminative methods because of the prior knowledge of the atoms' connectivity encoded in $G$ (i.e., a priori knowledge of chiral centers). In Diffusion settings, the generation starts from uninformative priors ($G_t, C_t$ are drawn from uninformative distributions at $t=1$). Conditional generation through classifier guidance would require a model $p_\phi(y|C_t, G_t), t\in [0,1]$, where $C_t$ and $G_t$ are uninformative for $t=1$, which is a highly non-trivial task for $p_\phi$ rotationally invariant and chiral-aware.
> > >
> > >
> > > Thank you for your thoughtful feedback and suggestions that have helped us illuminate some salient aspects of the problem setting, as well as reinforce the strengths of this work. We hope that your concerns have been satisfactorily addressed, and we would be greatful if the same will be reflected in stronger support from you for this work. Otherwise, we remain committed to engaging further and addressing any further concerns and suggestions.
> > >
> > >
> > > [1] Han Huang et al., "Learning Joint 2-D and 3-D Graph Diffusion Models for Complete Molecule Generation. IEEE, vol. 35, 2024.
> > >
> > > [2] Tuan Le et al., "Navigating the Design Space of Equivariant Diffusion-Based Generative Models for De Novo 3D Molecule Generation". ICLR 2024.
> > >
> > > [3] Ross Irwin et al., "Efficient 3D Molecular Generation with Flow Matching and Scale Optimal Transport". ICML 2024.
> > >
> > > [4] Hoogeboom, Emiel et al., "Equivariant Diffusion for Molecule Generation in 3D". PMLR 2022.
> > >
> > > [5] Clement Vignac et al., "MiDi: Mixed Graph and 3D Denoising Diffusion for Molecule Generation". ICLR, MLDD workshop, 2023.
> > >
> > > [6] Xingang Peng, et al. "MolDiff: Addressing the atom-bond inconsistency problem in 3D molecule diffusion generation". In ICML, 2023.
> > >
> > > [7] Lemeng Wu et al. "Diffusion-based Molecule Generation with Informative Prior Bridges". NeurIPS 2022
> > >
> > > [8] Bowen Jing et al. "Torsional diffusion for molecular conformer generation", NeurIPS, 2022.
> > >
> > > [9] Minkai Xu et al. "Geometric latent diffusion models for 3D molecule generation.". ICML, 2023.
> > >
> > > [10] Pedro Pinheiro et al. "3d molecule generation by denoising voxel grids". NeurIPS, 2023.

---

> > > > ### Comment · Reviewer_hRsL · 2024-11-20
> > > > **Preliminary comment on the response**
> > > >
> > > > I appreciate your comprehensive answer. I am providing a quick and preliminary response, allowing you to reiterate before the deadline. I agree that when I first reviewed the paper, I may have been a bit harsh regarding the importance of equivariance preservation, symmetry awareness, and the theoretical incorporation of chirality into equivariant neural networks. Your standalone approach of using fields as a representation and deep theoretical development makes the method quite unique and interesting.
> > > >
> > > > However, there are still a few things that seem odd to me:
> > > >
> > > > Neutrality as a Metric:
> > > >
> > > > Based on the data you provided for GEOM Drugs, 14% of molecules have at least one charged atom, while 86% of molecules have all atoms neutral. The metric indicates that JODO is capable of generating more neutral molecules than other approaches but is still far from perfect (14-86 I assume perfect because it is data distribution). Importantly, there's nothing chemically wrong with these 14% of molecules; having some charged atoms doesn't make a molecule unstable or undesirable. Therefore, using neutrality as a metric might not accurately reflect the quality of the generated molecules.
> > > >
> > > > Molecular Stability Metrics:
> > > >
> > > > Your comments on molecular stability are relevant. Different papers have used this metric in various ways, and generating simple bonds isn't the biggest issue—for example, EQGAT-Diff shows good fingerprint similarity with the test dataset, indicating that it's not generating trivial molecules. However, previous papers used some strange techniques to pump the metric: JODO uses RDKit functions that implicitly post-process molecules, and EQGAT-Diff and MiDi uses a somewhat unconventional lookup table to compute stability. However, the overall idea of using a metric that checks the valency of atoms is much stronger than the neutrality concept. If all neutral carbons in your dataset have a valency of 4, and in a generated molecule, carbon has a valency of 3 or 5, there's something strange with the molecule. These ground truth valency values should be collected from the dataset you're training on, and using a lookup table with charges is an overall relevant idea, because depending on the formal charge, the atom can exhibit different valency. I agree that the metric should be applied to raw model predictions without any explicit or implicit post-processing (like sanitizing) to evaluate the local quality of generated topologies genuinely.
> > > >
> > > > Use of MMFF Energy Distribution Metric:
> > > >
> > > > Using the MMFF energy distribution as a metric is quite weak because the structures for both QM9 and Drugs come from GEOM, which utilized CREST software with subsequent GFN2-xTB optimization. Each molecule is a local minimum of the potential energy landscape with respect to the GFN2-xTB energy function. All GEOM conformers have relative energy variability within 2.5 kcal/mol, which is a thermodynamically relevant interval (as noted in the original GEOM paper). MMFF is a much simpler approximation of the energy landscape, with an error of around 20 kcal/mol relative to GFN2-xTB. It would be much better to compare energy distributions using the same energy function used to generate the conformers.
> > > >
> > > > GEOM:
> > > >
> > > > It is inaccurate to refer to GEOM Drugs as GEOM because GEOM has two parts: QM9 and Drugs. If you are following EDM benchmarking, you should be using just GEOM Drugs for what you are calling GEOM. Also, the validity of 89% for GEOM Drugs does not seem to be accurate. As far as I remember, it should be above 99% or even 100%. Subsequently, the validity of JODO also appears to be 10% less compared to reported numbers.
> > > >
> > > > Axelrod S, Gomez-Bombarelli R. GEOM, energy-annotated molecular conformations for property prediction and molecular generation. Scientific Data. 2022 Apr 21;9(1):185.

---

> > > > > ### Author Response · Authors · 2024-11-24
> > > > >
> > > > > We thank the reviewer for all the effort put into this review, and their quick response. We address the raised points below.
> > > > >
> > > > > **Point 1. Neutrality as a Metric:** Based on the data you provided for GEOM Drugs, 14\% of molecules have at least one charged atom, while 86\% of molecules have all atoms neutral. The metric indicates that JODO is capable of generating more neutral molecules than other approaches but is still far from perfect (14-86 I assume perfect because it is data distribution). Importantly, there's nothing chemically wrong with these 14\% of molecules; having some charged atoms doesn't make a molecule unstable or undesirable. Therefore, using neutrality as a metric might not accurately reflect the quality of the generated molecules.
> > > > >
> > > > > *Non-neutral molecules being wrong:* we agree with reviewer that "neutrality" may create the confusing impression that non-neutral molecules are wrong, in general, and therefore explain in the text (Section 5, Basic molecule properties): "We aim to match the data neutrality, not to achieve 100\% neutrality". Please refer to Point 2 for a comparison between "stability" and "neutrality", and our motivation for chosing the latter.
> > > > >
> > > > > *Jodo generates more neutral GEOM-DRUGS:* We agree that metrics for JODO are better on GEOM-Drugs, but the comparison should account for the points made in "Weakness 1" of our previous rebuttal comment, namely: "Differences in GEOM benchmark" - using only the lowest energy conformation (JODO) results in roughly 23.5 times lower number of conformations compared to our training setting (using 30 lowest energy conformations, when available) and "Neutrality metric for aromatic methods compared to Kekule (ours)", which differs based on correct valencies considered (i.e., because of explicit aromatic bonds). The large number of different settings chosen by other methods, although perfectly valid, make strict comparisons between methods particularly challenging on GEOM-Drugs.
> > > > >
> > > > > **Point 2. Molecular Stability Metrics:** Your comments on molecular stability are relevant. Different papers have used this metric in various ways, and generating simple bonds isn't the biggest issue—for example, EQGAT-Diff shows good fingerprint similarity with the test dataset, indicating that it's not generating trivial molecules. However, previous papers used some strange techniques to pump the metric: JODO uses RDKit functions that implicitly post-process molecules, and EQGAT-Diff and MiDi uses a somewhat unconventional lookup table to compute stability. However, the overall idea of using a metric that checks the valency of atoms is much stronger than the neutrality concept. If all neutral carbons in your dataset have a valency of 4, and in a generated molecule, carbon has a valency of 3 or 5, there's something strange with the molecule. These ground truth valency values should be collected from the dataset you're training on, and using a lookup table with charges is an overall relevant idea, because depending on the formal charge, the atom can exhibit different valency. I agree that the metric should be applied to raw model predictions without any explicit or implicit post-processing (like sanitizing) to evaluate the local quality of generated topologies genuinely.
> > > > >
> > > > > We certainly agree that lookup tables are relevant ideas, but should be used cautiously.
> > > > >
> > > > > *Cases not identified by lookup tables:* If the data contains a heavily skewed distribution (e.g., C with valencies 3 and 5 are much less likely than C with valency 4), and a model generates C with those 3 valencies with equal probability, the lookup table (as employed in most recent works) will not capture this problem. Such an issue is quite striking, in particular, on QM9 - charged atoms are only present in 0.4\% of the data, while models still generate more than 10\% of molecules with charged atoms (our method generates 8.5\%). We deem this a very likely indication of generated molecules being unstable.
> > > > >
> > > > > *Stability metric:* Ideally, such a metric would account for much more than just valency and consider many chemical properties, while the "true" stability of a molecule cannot be assessed, short of synthesizing the compound in a lab. We agree with the reviewer that lookup tables may be relevant, but we argue that they should at least be employed using distribution distances, s.t. one can identify whether too many generated atoms are charged, which reveals suboptimal modeling and provides indications that significant proportions of molecules may be unstable. Instead of defining yet another metric, we chose to re-use "neutrality" (introduced in [1] with the name "stability") which does not have the problem explained in "Cases not identified by lookup tables".
> > > > >
> > > > > **REFERENCES**
> > > > >
> > > > > [1] Hoogeboom, Emiel et al., "Equivariant Diffusion for Molecule Generation in 3D". PMLR 2022.

---

> > > > > > ### Author Response · Authors · 2024-11-24
> > > > > >
> > > > > > **Point 3. Use of MMFF Energy Distribution Metric:** Using the MMFF energy distribution as a metric is quite weak because the structures for both QM9 and Drugs come from GEOM, which utilized CREST software with subsequent GFN2-xTB optimization. Each molecule is a local minimum of the potential energy landscape with respect to the GFN2-xTB energy function. All GEOM conformers have relative energy variability within 2.5kcal/mol, which is a thermodynamically relevant interval (as noted in the original GEOM paper). MMFF is a much simpler approximation of the energy landscape, with an error of around 20kcal/mol relative to GFN2-xTB. It would be much better to compare energy distributions using the same energy function used to generate the conformers.
> > > > > >
> > > > > >
> > > > > > **Note: Table 4 (Appendix A.1) has been split into Tables 4 and 5 in the Appendix A.1, in order to fit CE$\_{xTB}$.**
> > > > > >
> > > > > > **Note 2: histograms of xTB generated energies are reported in Figures 21 and 22, in Appendix A.1 (not respecting fig. counter s.t. rebuttal references are consistent - will change for camera-ready).**
> > > > > >
> > > > > > Thank you for the suggested metric. We now compute two versions of CE W$\_1$ distances, CE$\_xTB$ and CE$\_{MMFF}$, accounting for outliers.
> > > > > >
> > > > > > *MMFF motivation:* Although MMFF has a larger error, it explicitly accounts for covalent bond types in its computation. Therefore, we find it relevant, as most methods (e.g., all except [1] in the table below) explicitly generate bonds.
> > > > > >
> > > > > > | Model         | TV$\_a$     | QM9 CE$_{xTB}$     | QM9 CE$\_{MMFF}$   | GEOM-Drugs CE$\_{xTB}$ | GEOM-Drugs CE$\_{MMFF}$ |
> > > > > > |---------------|--------------------|--------------------------|--------------------------|-----------------------------|-----------------------------|
> > > > > > | EDM           | 0.49              | 0.30                    | 3.45                    | 1.93                       | -                           |
> > > > > > | MiDi          | 0.19              | 0.34                    | 3.12                    | 1.82                       | 40.98                      |
> > > > > > | JODO          | 0.19              | 0.22                    | 1.41                    | 1.21                       | 7.59                       |
> > > > > > | FMG           | 0.52± 0.02        | 0.71 ± 0.02             | 2.15 ± 0.385            | 3.22                       | 53.14                      |
> > > > > > | FMG$\_{β=0}$ | 1.02              | 0.98                    | -                        | -                          | -                          |
> > > > > >
> > > > > > *Discussion on the results:* We include the following in Appendix A1: Fields cannot enforce the number of atoms ($N$) that will be generated, while point clouds do so, sampling $N\sim p\_\mathcal{D}(N)$ during testing. While not enforcing the number of atoms provides useful flexibility (e.g., in conditional generation, the conditioning may be $p(\cdot| y)$, instead of $p(\cdot| y, N)$, where the latter requires knowledge of the joint $p(y,N)$), we find TV$\_a$ (ablation Table 3) and CE$\_{xTB}$ (Table 4) to gain significant improvements as a result of atom count conditioning (from no conditioning, FMG$\_{\beta=0}$, to FMG). However, in this work, we opted for classifier-free guidance on binned atom intervals (see Appendix G), rather than conditioning on specific atom counts. The high impact of atom count conditioning on these metrics provides a strong indication that, if more fine-grained conditioning is adopted, fields will achieve similar TV$\_a$ and CE$\_{xTB}$ numbers as point-clouds.
> > > > > >
> > > > > > **Point 4. GEOM** It is inaccurate to refer to GEOM Drugs as GEOM because GEOM has two parts: QM9 and Drugs. If you are following EDM benchmarking, you should be using just GEOM Drugs for what you are calling GEOM. Also, the validity of 89 for GEOM Drugs does not seem to be accurate. As far as I remember, it should be above 99\% or even 100\%. Subsequently, the validity of JODO also appears to be 10\% less compared to reported numbers.
> > > > > >
> > > > > > *GEOM name:* Thank you - we used GEOM-Drugs and changed the name to "GEOM-Drugs" to disambiguate.
> > > > > >
> > > > > > *Validity number:* We compute validity as the percentage of RDKit-parseable molecules, mainly checking if molecules can be kekulized and if atoms' valencies are $\le$ neutral valency, as defined in [1]. Different than [1], we include H atoms in validity computation. We confirm that most GEOM-Drugs molecules are valid (99.99\%) if we assign the RDKit molecule's atoms charges from molecular graphs for C, N, and O atoms.
> > > > > >
> > > > > > *Including H in validity*: In explicit aromatic settings (e.g., MiDi, JODO), aromatic rings need H atoms specified, otherwise Kekulization fails (e.g., pyrrole needs the N's H explicit).
> > > > > >
> > > > > > We thank the reviewer for their very timely response to our rebuttal. We are grateful for reviewer's attention to details, and hope to have addressed the additioinal points, which further enhance our work's rigurousness  and benchmark completeness. With the discussion deadline approaching, we hope the reviewer will consider updating their initial grade.

---

> > > > > > > ### Comment · Reviewer_hRsL · 2024-11-28
> > > > > > > **Rebuttle Response**
> > > > > > >
> > > > > > > Thank you for adding the xTB energy distribution and providing comprehensive answers to my questions. I have increased my scores primarily because of the interesting method you've presented. However, I still believe that using neutrality as a metric can be misleading.

---

> > > > > > > > ### Author Response · Authors · 2024-11-29
> > > > > > > >
> > > > > > > > Your concern about potential misinterpretation of neutrality (defined in [1,2,3] as "stability") is valid and highlights the importance of precise communication regarding its role in molecule generation evaluation. By no means, do we advocate neutrality as a direct measure of overall molecular stability; rather, it provides insight into the charge distribution characteristics of the generated molecules.
> > > > > > > >
> > > > > > > > Acting on your concern, we suggest treating neutrality as any other molecular distribution property (e.g., atom and covalent bond type distributions, xTB energies) in that the generated molecules should match the data distribution. On one hand, it is consistent with the general objective for generative modeling (generate molecules such that their distribution resembles the training distribution for each such property). On the other hand, it captures an important feature of chemical plausibility that helps mitigate the issue that molecular stability cannot unequivocally be assessed without synthesizing the molecules (and lookup-table-based stability seems to significantly overestimate stability, based on the charged molecule discrepancy we discovered in the experiments as we reported in Point 2).
> > > > > > > >
> > > > > > > > In the revised version, we will thus clearly disambiguate how "neutrality" should be interpreted in our evaluation metric description, thereby circumventing it from being misconstrued or overstated. We hope this clearly addresses your concern. We’re grateful for your feedback that has helped consolidate the contributions and the key takeaway messages of this work, and we shall greatly appreciate if you could revisit your score to reflect the same (and that you like the paper in general). Many thanks!
> > > > > > > >
> > > > > > > > [1] Hoogeboom, Emiel et al., "Equivariant Diffusion for Molecule Generation in 3D". PMLR 2022.
> > > > > > > >
> > > > > > > > [2] Lemeng Wu et al. "Diffusion-based Molecule Generation with Informative Prior Bridges". NeurIPS 2022
> > > > > > > >
> > > > > > > > [3] Minkai Xu et al. "Geometric latent diffusion models for 3D molecule generation.". ICML, 2023.

---

### Official Review · Reviewer_QxY8 · 2024-11-02

**Soundness:** 3
**Presentation:** 3
**Contribution:** 3
**Rating:** 8
**Confidence:** 4

**Summary:**

The authors identify a critical limitation in current E(3)-equivariant models for 3D molecular generation: they cannot properly account for molecular chirality, which is crucial for drug efficacy and safety. To address this, they introduce a field-based representation approach that replaces rotational symmetry constraints with reference rotations, allowing their model to correctly capture chiral properties while maintaining competitive performance with existing E(3)-based methods. They provide theoretical proofs showing why E(3)-equivariant models must necessarily ignore chirality, and demonstrate empirically that their method (FMG) achieves state-of-the-art performance in molecular neutrality while successfully capturing chiral properties, though at the cost of increased computational complexity for larger molecules.

**Strengths:**

The work identifies and addresses a fundamental limitation in current E(3)-equivariant models regarding chirality, which has been largely overlooked in previous research. Besides, it provides original theoretical proofs about the inherent limitations of E(3)-equivariant models in handling chirality.

The overall quality is high, reflected by the rigorous theoretical foundation with formal proofs about E(3) invariance and chirality (Propositions 1-3) and comprehensive empirical validation on standard benchmarks (QM9 and GEOM datasets). Well-structured presentation with clear motivation and problem statement. Ablation studies and

**Weaknesses:**

The first weakness is the method's computational complexity scales with the volume of the 3D grid (H×W×D), making it potentially intractable for larger molecules. As the author mentioned chirality in amino acids, will be a bottleneck for implementing the model in bio- and pharma-tasks. The current implementation requires high-resolution grids to maintain accuracy, leading to significant memory requirements. Another limitation is the analysis of field parameter sensitivity (σ, γ) is not very clear.

**Questions:**

1. Could adaptive grid resolutions or sparse representations help reduce memory requirements while maintaining accuracy?
2. How sensitive is the model's performance to the choice of field parameters (σ, γ)? What's the theoretical basis for choosing these values?
3. Why did the authors choose 0.33Å resolution for the grid, and how would different resolutions affect the quality-efficiency trade-off?
4. How might this approach be extended to other molecular properties beyond chirality?
5. Since chirality is a crucial property of molecules that are involved in drug discovery, is there any specific case that the author can provide on the FMG? Other than that, some properties that are not related to the chirality of the molecule (i.e. orbital energies), why FMG outperforms?
6. Can this model be utilized in total synthesis and retrosynthesis?

---

> ### Author Response · Authors · 2024-11-19
>
> We thank the reviewer for their time and effort in reviewing our work. Below, we address the raised questions.
>
> **Weakness 1.1** The first weakness is the method's computational complexity scales with the volume of the 3D grid (H×W×D), making it potentially intractable for larger molecules. As the author mentioned chirality in amino acids, will be a bottleneck for implementing the model in bio- and pharma-tasks. The current implementation requires high-resolution grids to maintain accuracy, leading to significant memory requirements.
>
>
> We agree with the reviewer that this is an important aspect to consider. We renamed Section 4.5 to "*Architecture choice, rotation invariance, and scaling*" to highlight the complexity analysis of our method. Appendix J now also contains an analysis of the memory overhead, where we compare favorably on large batch sizes (please see "General rebuttal", "Memory scaling" and Appendix J, Figure 20). We also note the "Question 3" response, "Resolution" parameter, and the new ablation we include in Appendix C - these show that utilizing lower resolution may be possible, which would allow larger molecules to be fit by the model. Still, whole proteins are unlikely to work as fields, but conditioning on "pockets" could be feasible.
>
> **Question 1** Could adaptive grid resolutions or sparse representations help reduce memory requirements while maintaining accuracy?
>
> Interesting ideas! We believe both approaches would likely require different architecture parameterizations for viability. We briefly experimented with DiT [1] early on in the method development, and compared it against UNet, but used the same atom $\pmb{u}\_a$ and bond $\pmb{u}\_b$ tensor fields for both, and therefore had to restrict to a shallow Transformer because of scaling constraints ($|\mathcal{G}|^2$ scaling, with $\mathcal{G}$ being the evaluated points). Utilizing adaptive resolutions/sparse representations may, however, bring considerable benefits and is likely a promising future avenue.
>
> [1] Peebles et al., "Scalable Diffusion Models with Transformers". IEEE/CVF 2023.
>
> **Question 2** How sensitive is the model's performance to the choice of field parameters ($\sigma$, $\gamma$)? What's the theoretical basis for choosing these values?/ **Weakness 1.2** Another limitation is the analysis of field parameter sensitivity ($\sigma$, $\gamma$) is not very clear./ **Question 3** Why did the authors choose 0.33Å resolution for the grid, and how would different resolutions affect the quality-efficiency trade-off?
>
>
> Thank you for noticing this. Field parameter analysis should prove useful for future development: we now add ablations of field parameters in Appendix C (Figures 8 and 9). The following were added in Appendix C:
>
> The field hyperparameters were selected during early experimentation. We focused on the effect of field hyperparameters on fitting $u\_a(x), u\_b(x)$ to the generated fields $\pmb{u}\_{a},\pmb{u}\_{b}$ (Equations 13 and 14 and Appendix E). We compute the "extraction accuracy", which measures how many molecules have the same molecular graph extracted from the fields they create. We additionally report the average "distance from original", computed as the average Euclidean distance between the correct atom positions and the ones extracted. For both, we determine how robust they are for various levels of noise applied to the fields. Ablation of field hyperparameters for the resulting diffusion model's performance was not performed, due to heavy computational resources required.
>
> *Field component standard deviation* $\sigma$: Intuitively, larger standard deviations $\sigma$ would facilitate smaller receptive field CNN kernels to capture longer distance neighboring RBF components. At the same time, we require that molecular graphs and conformations are correctly extracted. Figure 8 shows how increasing $\sigma$ and $r$ worsens the noise robustness. However, the more coarse resolution of $0.44$ would allow for larger architectures. Since it is sensible to assume larger architectures will produce less noise, this option is very sensible for future development.
>
> *Field component weight* $\gamma$: The component weights may be thought of as input an input normalization parameter, and we therefore did not consider other values that $0.176$, which should normalize input to (roughly), values between $[0,1]$. We believe this parameter is unlikely to change the model's performance.
>
> *Resolution* $r$: In general, it is desired to have as low resolution as possible, while ensuring that conformer and graphs can still be extracted with sufficient accuracy. Figure 9 shows how lowering the resolution from 0.33 to 0.44Å worsens "extraction accuracy" and "distance from original" across noise levels. We note, however, that $r=0.44$ may still be a valid choice, as it would make larger NN parametrizations feasible, and therefore the field noise levels are likely going to be smaller.

---

> > ### Author Response · Authors · 2024-11-19
> >
> > **Question 4** How might this approach be extended to other molecular properties beyond chirality?
> >
> > Thank you for your question. We conducted experiments on molecule property conditional generation in Appendix A.2 (Table 6). Since the method now can principally model all molecular distribution properties, conditional generation could be similarly employed for any property.
> >
> > **Question 5.1** Since chirality is a crucial property of molecules that are involved in drug discovery, is there any specific case that the author can provide on the FMG?
> >
> > The main experiment highlighting the chiral-unaware modeling drawbacks is the chirality analysis in Section 6.4, which empirically confirms our theoretical result on E(3) invariances and chirality (enantiomer pairs are not distinguishable in for E(3) invariant methods) while showing that our method captures the skewed enantiomer pair. It remains practically difficult to show this outside a controlled setting, but we are excited about the prospects of e.g., conditional generation that requires chiral-aware modeling.
> >
> > **Question 5.2** Other than that, some properties that are not related to the chirality of the molecule (i.e. orbital energies), why FMG outperform?
> >
> > We do not report orbital energies, but instead report distribution distances between generated and data Merck Molecular Force Field (MMFF). Our better result on QM9 is likely a consequence of our field representation's flexibility. Namely, as we do not enforce the generation of a strict number of atoms, the model can remove or add atoms freely during generation, which probably leads to more "natural" bond lengths and angles. Our better performance on bond lengths (BL), together with higher "neutrality" (valency being better) are indications of that.
> >
> > **Question 6** Can this model be utilized in total synthesis and retrosynthesis?
> >
> > Thank you for the question. These problems fall slightly outside the current work's scope, but we think these are exciting future directions! Our method can be in principle used to explore new compounds that respect certain chemical properties, and some could be better suited for retrosynthesis than known molecules. However, additional tools are likely necessary to create a full pipeline and aid in describing potential pathways resulting in the generated molecules.
> >
> > Thank you for your thoughtful feedback, and questions, and for helping us strengthen our work with additional experiments. We hope the questions raised have been satisfactorily answered. Otherwise, we remain committed to engaging further and addressing any further questions and suggestions.

---

> > > ### Comment · Reviewer_QxY8 · 2024-11-25
> > >
> > > After carfully reading the author's reply and other reviews comment, I will maintain my rate (8: Accept) and I appreciate the author's detailed and insightful replies. Interesting work!

---

### Official Review · Reviewer_V7Qk · 2024-11-04

**Soundness:** 2
**Presentation:** 3
**Contribution:** 3
**Rating:** 6
**Confidence:** 3

**Summary:**

This paper proposes to generate molecules by representing them as functions in 3D space. Each atom is represented as a gaussian distribution centered at the atom position. The bonds are represented similarly with the center of the gaussian being placed at the center of the bonds. Compared to  such a representation can learn the chirality since it bears the reflection invariance. Experiments are conducted on QM9 and GEOM.

**Strengths:**

- Since most deep learning methods use GNNs for molecules, using CNNs is novel.
- As stated by the authors, capturing chirality is important for representing molecules. And GNN-based methods may have difficulty in breaking  the reflections symmetry.

**Weaknesses:**

- Although alignment methods are employed, the proposed method cannot guarantee the rotation invariance.
- As shown in table 2, the molecule graph quality metric is not good on the QM9 dataset.

**Questions:**

- What's the trade-off between losing rotation invariance and gaining the ability of learning chirality?

---

> ### Author Response · Authors · 2024-11-19
>
> We thank the reviewer for their time and effort in reviewing our work. Below, we address the raised questions.
>
>
> **Weakness 1** Although alignment methods are employed, the proposed method cannot guarantee the rotation invariance.
>
>
> Irrespective of an initial molecule's orientation, the algorithm always deterministically rotates it s.t., the largest variation axes are aligned (in this order) on the height, width, and depth. Compared to previous methods that use E(3) invariant NN and can generate molecules with any rotation, our method (on optimality) will only generate molecules according to this frame of reference.
>
> The specific orientation choice is related to the NN's generalization. Ideally, similar molecules will have similar orientations, resulting in their field densities being aligned well. Intuitively, aligning molecules on the highest axes variance would place various conformer categories (e.g., long and planar molecules, respectively), in roughly the same orientations, allowing our $\theta$  NN's activations to be similar for similar molecules. The degree to which this is true will determine how efficiently our NN's parameters are used, and in turn, the performance, (e.g, Table 3, using rotation augmentation FMG$\_{aug}$'s 82.6 Mol neutrality increases to FMG$\_{\beta=2}$'s 91.5, when the reference rotations in Section 3.5 are used).
>
>
> **Weakness 2** As shown in table 2, the molecule graph quality metric is not good on the QM9 dataset.
>
>
> We argue that relative to other methods, our method maintains competitive MST and bond type distribution TV$\_b$. Our TV$\_a$ performance is sub-optimal, but we explain how this is likely a result of our model trading this property in favor of neutral molecule generation, which is not possible for point-cloud methods, as they are hard-constrained to generate a pre-specified number of atoms. Our ablation study provides further indication for this hypothesis, since, in Table 3, we obtain significant improvement on TV$\_a$, as a result of adding atom number conditioning $N$ (from no conditioning, FMG$\_{\beta=0}$ to FMG$\_{\beta=2}$).
>
>
> **Question 1** What's the trade-off between losing rotation invariance and gaining the ability of learning chirality?
>
> Chirality is one 3D property that influences the physical, chemical, and biological properties of molecules, similar to all other isomeric molecular properties. Therefore, modeling molecules with E(3) invariance will always be limited. We argue that chirality awareness is especially important in organic chemistry, as enzymes/proteins are chiral, and often skew the distribution of optimal and safe drugs. This motivated us to explore the limitations of invariance groups wrt. chirality, proving the lack of chiral awareness in E(3), and further exploring computational constraints of adapting E(3) to SE(3), which is prohibitive ($\mathcal{O}(N^4)$).
>
>
> In novel compound generation (joint generation of molecule and conformer graphs $p(C,G)$), rotational invariance is stated as a strict requirement for good generalization. However, we argue that non-invariant methods may still be employed successfully (e.g, Table 3, using rotation augmentation FMG$\_{aug}$'s 82.6 Mol neutrality increases to FMG$\_{\beta=2}$'s 91.5, when the reference rotations in Section 3.5 are used). Further evidence of this is shown in a parallel line of work that considers the generation of $p(C|G)$, where results suggest that good performance can be achieved without restricting the NN parametrizations [1,2].
>
> [1] Yuyang Wang et al., "Swallowing the Bitter Pill: Simplified Scalable Conformer Generation". ICML 2024.
>
> [2] Josh Abramson et al., "Accurate structure prediction of biomolecular interactions with AlphaFold 3". Nature 2024.
>
> Thank you for your thoughtful feedback. We hope that your concerns have been satisfactorily addressed, and we would be greatful if the same will be reflected in stronger support from you for this work. Otherwise, we hope to further engage and address further concerns and suggestions.

---

> > ### Comment · Reviewer_V7Qk · 2024-11-26
> >
> > Thank you for the clarifications. I agree that the model is rotation-invariant. A better question might be the robustness of rotation invariance, i.e., how well the invariance is kept approximately when the molecular structure is perturbed (e.g., a bond is stretched slightly). However, I think this is a common challenge faced by canonicalization methods and may be hard to solve in general. For evaluation, although I am not very familiar with the specific metrics, I think the explanation for the tradeoff in the experimental results is helpful. I still feel this method is an interesting contribution. I will keep my rating.

---

### Official Review · Reviewer_Gfde · 2024-11-04

**Soundness:** 3
**Presentation:** 4
**Contribution:** 3
**Rating:** 8
**Confidence:** 3

**Summary:**

This paper tackles the limitations of E(3)-invariant features in modeling chirality, as mirror symmetry is inherently implied in such features. The authors first provide a theoretical motivation, illustrating why E(3)-invariant features are inadequate for representing chirality. They then propose using fields to model molecular graphs, learning these fields through a reverse diffusion process. The effectiveness of their approach is benchmarked on the QM9 and Geom datasets.

**Strengths:**

Chiral awareness in E(3)-invariant architectures is essential for accurately modeling molecular properties. I appreciate the authors’ approach of representing molecules as collections of fields corresponding to atoms and bonds. These fields, as probability densities, closely mirror the quantum mechanics formalism than point clouds, where molecules are wavefunctions. This aligns with methods like Density Functional Theory (DFT), which models charge density and provides a more expressive representation than point clouds. However, the impact on memory overhead from this approach warrants further investigation. Overall, I found this paper to be well-written, clearly delineating previous work, the contributions of this study, and its limitations.

**Weaknesses:**

The impact of this approach on memory overhead requires further investigation. How does the memory overhead compare to point cloud representations? Could you provide insights on how memory usage scales with increasing molecule size and greater diversity of atom and bond types?

**Questions:**

Please see weakness.

---

> ### Author Response · Authors · 2024-11-19
>
> We thank the reviewer for their time and effort in reviewing our work. Below, we address the raised questions.
>
> **Weakness 1** The impact of this approach on memory overhead requires further investigation. How does the memory overhead compare to point cloud representations? Could you provide insights on how memory usage scales with increasing molecule size and greater diversity of atom and bond types?
>
>
> We agree with the reviewer that this is an important aspect to consider. We renamed Section 4.5 to *"Architecture choice, rotation invariance, and scaling"* to highlight our complexity analysis, which now includes the memory overhead. Please refer to the "*General Rebuttal*", section "*Scalability*" for a short explanation and Appendix J. Appendix J for an in-depth analysis.
>
> Thank you for your suggested analysis and helping us improve our manuscript. We hope our new experiment has addressed the memory overhead aspects, and hope that you can provide further suggestions if not.

---

> > ### Comment · Reviewer_Gfde · 2024-11-26
> > **Keeping the score.**
> >
> > Thank you for answering my questions. I will keep the score at 8.

---

### Author Response · Authors · 2024-11-19
**General Rebuttal**

We thank the reviewers for their time, valuable insights, and questions, and the AC and the senior AC for their service to the community.

We added the scalability analysis comparing point clouds with field representations here, as this point was repeatedly raised (section Scalability of "General rebuttal").

We addressed each comment in individual replies and modified the manuscript wherever necessary. Text additions are marked in blue and removed text with strikethrough. Please see the updated manuscript. We note a few changes in the reported results:


* JODO [1] has been included in all tables.
* We now account for outliers when computing BA, BL, and EC metrics$^{\pmb{1}}$ (Appendix A.1, Table 4), by removing all generated bonds and angles outside the dataset limits.

(**1**) Wasserstein distance W$_1$ is outlier sensitive. In principle, e.g., the W$_1$  bond length distance (BL), can become arbitrarily high based on high enough bond distance outliers. Our method has strict upper limits on bond lengths (defined by evaluated points $\mathcal{G}$), while point-cloud approaches do not. As such, we ensure the reported conformation metrics aren't better in our case due to a few outliers created by point-cloud methods by removing outliers from all methods. The results did not change significantly (Table 4).

# Scalability:
We renamed Section 4.5 to *"Architecture choice, rotation invariance, and scaling"*, and now briefly comment on the time and memory complexity, and refer to Appendix J for an in-depth analysis.

## Time scaling:

In short, point-cloud-based methods time scaling is $\mathcal{O}(N^2)$, while our method depends on the shapes of molecules in a dataset $\mathcal{D}$. We therefore derive a theoretical upper limit for our method's scaling based on the number of atoms $N$. Our result shows that, when molecules are chosen adversarially, our scaling is

$\frac{1.35N}{r} \cdot \frac{1.35N}{2r} \cdot \frac{1.35N}{3r},$


for some resolution $r$. In practice, however, we show in Appendix J that the method's scaling differs substantially from this upper limit, and we even obtain faster generation time than EDM on the largest GEOM molecules.


## Memory scaling:

Increasing the molecule sizes has a significant impact on memory overhead. Similar to the analysis done in the time scaling analysis in Appendix~J.0.1, we construct grids that can fit all GEOM molecules with a certain number of atoms N. We report the maximum memory allocated for EDM [2] and our method, when generating samples of various atom sizes (determined on the x label by the number of atoms $N$, or dots denoting volume sizes), with 32 samples per batch.

Figure 20 a) shows that our method performs better for larger molecules, but worse for smaller ones. We note here that GNN architecture batching is done by considering a single large graph, where nodes from different batches are disconnected. Therefore, adjacency matrices of this implementation have poor scaling for large batch sizes: $((\sum_{b\in B}{N_b})^2)$, where $B$ is the batch sizes and $N_b$ is the number of atoms in a molecule.

We find that increasing the number of unique channels (atoms or bonds) is negligible and linear. The additional memory required stems from the input CNN layer's kernel dimension (the depth sizes of the initial 3D CNN kernels are equal to $|\mathcal{A}| + |\mathcal{B}|$), and the larger intermediate states. However, these are considerably smaller than the architecture's overhead (Figure 20 b). Channel scaling is performed with a batch size of 1.

[1] H. Huang et al., "Learning Joint 2-D and 3-D Graph Diffusion Models for Complete Molecule Generation". IEEE Transactions on Neural Networks and Learning Systems, Sept. 2024.

[2] Hoogeboom, Emiel et al., "Equivariant Diffusion for Molecule Generation in 3D". PMLR 2022.

---

### Comment · Area_Chair_ezFc · 2024-11-22

Hi reviewers,

The authors have posted their rebuttals. If you haven't done so, could you please check their responses and engage in the discussions? Please also indicate if/how their responses change your opinions.

Thanks,

AC

---

### Meta-Review · Area_Chair_ezFc · 2024-12-20

**Metareview:**

This paper proposes a CNN-based method for molecular generation, focusing on effectively capturing chirality which usually fails to be
 incorporated in E(3) invariant features. The method represents molecules as functions in 3D space. Each atom is represented as a Gaussian distribution centered at the atom position. The method employs fields to generate model molecular graphs, learning these fields through a reverse diffusion process. Given that the chiral awareness in E(3)-invariant architectures is essential for molecular generation and the proposed method is technically novel and empirically strong, I recommend the acceptance of the paper.

**Additional Comments On Reviewer Discussion:**

Three reviewers among four are positive about the paper and most of their concerns have been addressed during the rebuttal. Reviewer V7Qk suggests including an additional experiment on the tradeoff in the next version of the paper. The 4th reviewer, Reviewer hRsL had a thorough discussion with the authors, and in her/his last comment, Reviewer hRsL generally agreed the method is interesting and finally increased the rating to 6.

---

### Decision · Program_Chairs · 2025-01-22

Accept (Poster)